# Reproducible analysis of disease space via principal components using the novel R package syndRomics

**Abel Torres-Espín[1,2,3], Austin Chou[1,2,3], J Russell Huie[1,2,3], Nikos Kyritsis[1,2,3], Pavan S Upadhyayula[4], Adam R Ferguson[1,2,3,5]***

[1]Weill Institute for Neurosciences, Brain and Spinal Injury Center (BASIC), University of California, San Francisco (UCSF), San Francisco, United States; [2]Department of Neurological Surgery, University of California San Francisco (UCSF), San Francisco, United States; [3]Zuckerberg San Francisco General Hospital and Trauma Center, San Francisco, United States; [4]School of Medicine, University of California San Diego (UCSD), San Diego, United States; [5]San Francisco VA Health Care System, San Francisco, United States

**Abstract** Biomedical data are usually analyzed at the univariate level, focused on a single primary outcome measure to provide insight into systems biology, complex disease states, and precision medicine opportunities. More broadly, these complex biological and disease states can be detected as common factors emerging from the relationships among measured variables using multivariate approaches. 'Syndromics' refers to an analytical framework for measuring disease states using principal component analysis and related multivariate statistics as primary tools for extracting underlying disease patterns. A key part of the syndromic workflow is the interpretation, the visualization, and the study of robustness of the main components that characterize the disease space. We present a new software package, *syndRomics*, an open-source R package with utility for component visualization, interpretation, and stability for syndromic analysis. We document the implementation of *syndRomics* and illustrate the use of the package in case studies of neurological trauma data.

*For correspondence: adam.ferguson@ucsf.edu

**Competing interests:** The authors declare that no competing interests exist.

## Introduction

The goal of the burgeoning field of precision medicine is to understand complex disease states and provide opportunities for deep patient phenotyping and highly targeted therapeutics. Precision medicine requires an understanding of multidimensional disease states. Yet, the analysis of biomedical data remains largely univariate, with response variables considered individually and reports involving several distinct analyses. This analytical approach limits our interpretation of the complexity of a disease by not considering the shared information across variables and potentially contributing to irreproducibility due to statistical limitations of multiple comparison testing. Understanding the full set of interrelated disease features through multivariate statistics is the goal of the growing domain of 'syndromics' (*Ferguson et al., 2011*). In particular, principal component analysis (PCA) and related multivariate statistics such as nonlinear PCA or factor analysis have been proposed as tools for extracting underlying factors or patterns (principal components [PCs]) reflecting disease states (*Ferguson et al., 2013*; *Haefeli et al., 2017a*; *Haefeli et al., 2017b*; *Kutcher et al., 2013*; *Nielson et al., 2014*; *Nielson et al., 2015*; *Panaretos et al., 2017*; *Rosenzweig et al., 2010*; *Rosenzweig et al., 2018*; *Rosenzweig et al., 2019*; *Zhang and Castelló, 2017*). There are several other multivariate methods that could be used for multivariate pattern detection: other ordination and dimension reduction techniques, cluster analysis, discrimination analysis, or the plethora of more

recent machine learning methods. The use of any of these methods has its advantages and pitfalls (*Everitt and Hothorn, 2011*). We focus on PCA as being one of the most widely used method for pattern detection. PCA is a multivariate statistical procedure that allows for the generation of new uncorrelated variables, called PCs, as a weighted combination of the original variables (*Abdi and Williams, 2010*; *Hotelling, 1933*; *Jolliffe and Cadima, 2016*). These components are ordered such that the first component explains the major source of variance in the data, the second component the second largest source of variance, etc. The extracted components reflect the interrelation between all the original variables or features, allowing for disease pattern detection, guiding in the interpretation of disease complex space and overcoming univariate analysis limitations.

Despite the extensive use of PCA in some subfields of biological research and the increasing use of PCA for disease pattern discovery, there is very limited information in the literature that can guide applied biomedical researchers about its implementation and interpretation. Here, we offer a practical guide to the application of PCA for the extraction of disease patterns that conform the disease space, with focus on reproducibility. By no means can we cover the extensive field of PCA in the present document. Rather, we aim to provide an introductory manual to extraction of reproducible disease patterns using multidimensional analytics, directed to biomedical researcher practitioners while pointing to additional relevant sources of information. We introduce a software package for the R programming language called *syndRomics*, implementing some of the tools described here. We will illustrate the analysis workflow and the use of the package in experimental data from case studies in neurotrauma.

The key steps in disease pattern detection by PCA are shown in *Figure 1*. The *syndRomics* package offers functionalities that aid in these steps, building on the extensive PCA framework developed by the R open-source community. The package implements a novel visualization tool, the syndromic plot first published by *Ferguson et al., 2013*, as well as functions to quickly generate two other publication-ready visualizations (a heatmap and a barmap). In addition, the package implements resampling strategies, providing data-driven approaches to analytical decision-making aimed to reduce researcher subjectivity and increase reproducibility. In particular, the package offers a

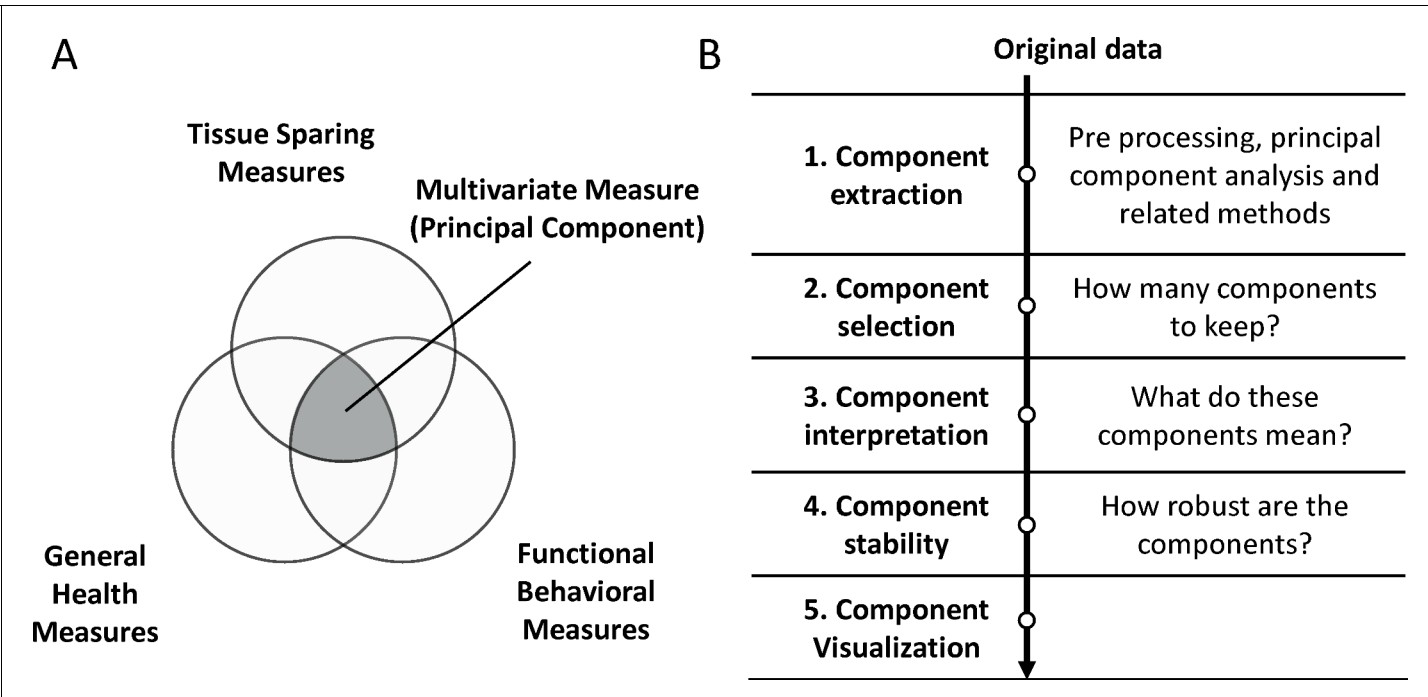

**Figure 1.** Summary of the syndromic framework and analysis steps. (**A**) The theoretical framework of syndromic analysis. The intersection between different outcome measures can create a multivariate measure (principal component if PCA is used) to explain different patterns of variance in the data. The conceptual union of three van diagram forms the core of the syndromic plot symbolizing the multidimensional measure. (**B**) The different steps of the workflow to using PCA such as for disease pattern analysis.

function to extract metrics for component and variable significance by using nonparametric permutation methods (*Landgrebe et al., 2002*; *Linting et al., 2011*; *Peres-Neto et al., 2003*), to inform component selection and component interpretation. Finally, the package incorporates functions to study component stability toward understanding the generalizability and robustness of the analysis (*Cattell and Baggaley, 1960*; *Cattell et al., 1969*; *Lorenzo-Seva and ten Berge, 2006*).

## Results

We will describe the general steps to use PCA for syndromic analysis and illustrate the use of the *syndRomic* package along the analytical steps with two case studies of neurotrauma data. Details of the usage and implementation of the package and functions are described in the Materials and methods section. The full code reproducing the analysis can be found in the supplementary material. Code boxes in the text provide snippets illustrating the main sections of the code. The first case study is used as a tutorial to illustrate the steps of analysis; the second case study is discussed at the end of the results section. For the first case, we used a publicly available preclinical dataset on the Open Data Commons for Spinal Cord Injury (odc-sci.org) (*Callahan et al., 2017*; *Fouad et al., 2019*). We selected a subset of the dataset with accession number ODC-SCI: 26 (*Ferguson et al., 2018*) that has been previously used for deriving the so-called spinal cord injury (SCI) syndromics (*Ferguson et al., 2013*). The dataset contains 159 subjects (rats) that have been studied on different motor functional outcomes across time after cervical spinal cord injury. The subset chosen for the present analysis consists of 18 outcome variables measured at 6 weeks after injury. The included variables for this analysis are shown in *Table 1*. For additional details of these variables, see *Ferguson et al., 2013*.

### Step 1: Extracting PCA solution from the data

There is extensive literature on performing PCA (*Abdi and Williams, 2010*; *Jolliffe and Cadima, 2016*; *Zhang and Castelló, 2017*). As a consideration, biomedical data aiming to capture the multivariate disease space usually contains variables of different types (i.e. categorical, continuous, etc.) and scales, known as 'mixed-type' data. Moreover, missing data is a common problem in biomedicine (*Hollestein and Carpenter, 2017*; *Kaushal, 2014*; *Nielson et al., 2020*) that needs to be solved

**Table 1.** List of variables included in the first case study.

| Variable | Definition |
| --- | --- |
| wtChng | Change of animal weight (grams) from day of Injury to 6 weeks post-injury |
| RFSL | CATWALK SYSTEM RightForelimb StrideLength at 6 weeks post-injury |
| LFSL | CATWALK SYSTEM LeftForelimb StrideLength at 6 weeks post-injury |
| RHSL | CATWALK SYSTEM RightHindlimb StrideLength at 6 weeks post-injury |
| LHSL | CATWALK SYSTEM LeftHindlimb StrideLength at 6 weeks post-injury |
| RFPA | CATWALK SYSTEM RightForelimb PrintArea at 6 weeks post-injury |
| LFPA | CATWALK SYSTEM LeftForelimb PrintArea at 6 weeks post-injury |
| RHPA | CATWALK SYSTEM RightHindlimb PrintArea at 6 weeks post-injury |
| LHPA | CATWALK SYSTEM LeftHindlimb PrintArea at 6 weeks post-injury |
| StepDistRF | CATWALK SYSTEM RightForelimb Step Distribution Deviation from 25% at 6 weeks post-injury |
| StepDistLF | CATWALK SYSTEM LeftForelimb Step Distribution Deviation from 25% at 6 weeks post-injury |
| StepDistRH | CATWALK SYSTEM RightHindlimb Step Distribution Deviation from 25% at 6 weeks post-injury |
| StepDistLH | CATWALK SYSTEM LeftHindlimb Step Distribution Deviation from 25% at 6 weeks post-injury |
| TotalSubscore | Total BBB Subscore at 6 weeks post-injury |
| BBB FergTrans | BBB Ferguson Transformation score 6 weeks post-injury |
| Groom | Grooming Score 6 weeks post-injury |
| PawPL | PawPlacement score 6 weeks post-injury |
| ForelimbOpenField | Forelimb openfield score at 6 weeks post-injury |

to be able to apply most standard PCA algorithms. Therefore, some pre-processing transformations are usually applied before performing PCA. For example, linear PCA is sensitive to the scale of variables, thus when applying a linear PCA to continuous variables of different units or scales, a common practice is to scale the data to unit variance first (i.e. equivalent to performing the PCA on the correlation matrix). The use of the package to conduct syndromics analysis from linear PCA is illustrated on the first case study. In cases of datasets with mixed data types and/or non-linear relationships between variables, nonlinear PCA with optimal scaling transformation (*Linting et al., 2007a*; *Mair and Leeuw, 2019*) has been previously used for disease pattern analysis (*Rosenzweig et al., 2018*; *Rosenzweig et al., 2019*). We used the syndRomics package to analyze patterns from a nonlinear PCA in the second case study. In cases with missing data, strategies such as data imputation or the use of PCA algorithms allowing missing values might be needed (*Dray and Josse, 2015*). While missing values analysis and dealing with missingness is an extensive topic that is not covered in detail here (*Rubin, 1976*; *Buuren, 2018*), the chosen case studies do contain missing values and illustrate how the package can help to determine the stability of the PCs when imputing missing values (see component stability section).

Another consideration is selecting which variables to include in the analysis. For PCA of experimental data where there are stratifying factors (e.g. control vs. treatment), it is important to leave out variables that directly capture the variance of these factors, which would bias PCA results toward separating the experimental groups. This bias is problematic since in syndromic analysis, the goal is to find the relationship between variables describing different diseases states in an unsupervised (i.e. not guided by our design) manner. For instance, if treatment indicators are included and the variance between treatment groups is high, the PCA solution would directly capture the experimental design and confound the multivariate patterns.

The disease components can be used in subsequent analysis as multivariate outcomes or predictor indicators (*Haefeli et al., 2017a*; *Nielson et al., 2015*; *Rosenzweig et al., 2018*; *Rosenzweig et al., 2019*). PCA is used to extract the correlation structure between variables, generating new independent variables as linear combinations. Beyond the use of PCs as proxies for disease patterns, the PCs can help mitigate issues that might appear when analyzing several variables such as multicollinearity, overfitting, and multiple testing (*Altman and Krzywinski, 2018*; *Johnson et al., 1973*; *Lever et al., 2017*).

The reader is referred to some materials of interest on considerations and limitations when conducting PCA and related methods for biomedical research (*Jiang and Eskridge, 2000*; *Konishi, 2015*; *Nguyen and Holmes, 2019*; *Zhang and Castelló, 2017*).

Case study: In the first case study, the goal is to run a linear PCA to study the motor function components 6 weeks after cervical spinal cord injury. This will summarize all motor function variables as a small set of independent components explaining different aspects of the motor behavior after an SCI. The data contains missing values (*Figure 4—figure supplement 1*), and therefore we performed missing values analysis before continuing with the workflow. Typically, the first step in missing values analysis is to determine patterns of missingness and classify missing values as missing completely at random (MCAR), missing at random (MAR) or missing not at random (MNAR) (*Rubin, 1976*). The type of missingness will guide the decision on which is an acceptable procedure to deal with missing values. For instance, deleting all subjects that contain at least one missing observation is common practice (aka listwise deletion or complete-case analysis), but it is only acceptable if missing values are MCAR. Otherwise, the robustness and proper estimation of the missing values can not be guaranteed (*Schafer and Graham, 2002*; *Buuren, 2018*). In the example data, subjects have been pooled together from different experiments. We know that the observed pattern of missingness (*Figure 4—figure supplement 1*) is due to a set of animals where some of the outcome measures were not studied, suggesting that missing values are MNAR. We confirmed that missing values are not MCAR using a previously described test of MCAR (*Jamshidian and Jalal, 2010*) implemented in the *MissMech* package in R (*Jamshidian et al., 2014*), which rejected the hypothesis of MCAR missingness in our data. Thus, excluding subjects from the analysis is not justified. Instead, we have used multiple imputation through the *mice* R package (*Buuren and Groothuis-Oudshoorn, 2011*) to generate 50 imputed datasets and pooled them using the mean of each observation. We will illustrate on the component stability section how the *syndRomics* package can be used to determine the robustness of multiple imputation for disease pattern analysis. We extracted the PCA solution of the pooled imputed data using the *prcomp()* function in R after

centering and scaling the data to unit variance (R code box 1). Other similar functions in R or other software can be used.

```
R Code Box 1
pca<-prcomp (pca_data, center = TRUE, scale. = TRUE).
```

## Step 2: Component selection: how many components to keep?

The first question, after running PCA for extracting the disease components is usually to determine how many PCs are relevant. As a general consideration, the PCs with lower eigenvalues (i.e. explain less variance) have a higher chance of representing noise in the data (*Jolliffe and Cadima, 2016*), questioning their generality and value. The goal is to determine the minimal set of components that can be used to describe the disease space. Importantly, there is not a single, specific rule for this determination. A common method in PCA and related methods is the Scree test by *Cattell, 1966*, where all PCs are ordered in descending rank by their eigenvalues, and PCs above the 'elbow' are retained. Another criterion is the eigenvalue greater than one rule which is applied to standardized PCAs (from the correlation matrix) with the criteria of only keeping PCs with an eigenvalue (i.e. the variance of a component) above 1 (*Guttman, 1954*; *Kaiser, 1960*). A more thorough description of these and others methods can be found elsewhere (*Glorfeld, 1995*; *Horn, 1965*; *Vitale et al., 2017*; *Zwick and Velicer, 1986*). Simulations have shown these methods (specially the eigenvalue greater than one rule) to be less robust than a re-sampling approach for selecting the number of relevant components (*Zwick and Velicer, 1986*). The *syndRomics* package incorporates a nonparametric permutation test approximated through Monte Carlo re-sampling of the total 'variance accounted for' (VAF) of each PC to aid in the selection of relevant PCs (*Buja and Eyuboglu, 1992*; *Glorfeld, 1995*; *Horn, 1965*; *Landgrebe et al., 2002*). The permutation test can also assist in component interpretation by studying the contribution of each variable to the PCA solution (*Buja and Eyuboglu, 1992*; *Linting et al., 2011*) as we will see in the next section.

The goal of the permutation test is to determine whether the extracted PCs can be considered to be generated not-at-random. This method has been shown to outperform parametric tests for PCA in situations similar to biomedical data where sample sizes are relatively small and the data rarely comply with the assumptions of the models (*Buja and Eyuboglu, 1992*; *Horn, 1965*; *Zwick and Velicer, 1986*). In that regard, a hypothesis test is defined as:

$H_{(null)}$:PC VAF is indistinguishable from a random generation
$H_{(alternative)}$:PC VAF is different from random

The p values are calculated by:

$$p = (q+1)/(P+1) \tag{1}$$

where $q$ is the number of times the chosen metric is higher in the permuted distribution than in the original PCA solution and $P$ is the number of permutations (*Buja and Eyuboglu, 1992*). Rejecting the null hypothesis is interpreted as evidence of the tested PC being generated from true signal and not by random noise. This sets a lower bound for which PCs to consider 'important' above noise, but does not indicate the magnitude of the 'importance', which is represented by VAF. Importantly, for datasets with several directions of variance and high signal-to-noise ratio, PCs with low VAF can still be statistically significant. The value of interpreting such PCs must be judged by the researcher in the context analysis in question. It is also important to consider how big $P$ needs to be when performing re-sampling, such as with the permutation test incorporated in the package. The reader should note that the lowest $p$ value that can be calculated is dependent on $P$. For example, if $P$ is set to a value of 10 (a relatively low value), the smallest p value that can be detected is 0.09, which occurs when $q = 0$. Accordingly, $P$ should be set high enough to reach the desired minimum p value. Moreover, simulation studies have shown that $P$ under 99 have low power and a minimum of 499 permutations is recommended (*Buja and Eyuboglu, 1992*; *Abdi and Williams, 2010*; *Linting, 2007*). By default, we have set the number of permutations to 1000 (smallest p value approximately equal to 0.001) as this has been shown to produce good results (*Landgrebe et al., 2002*; *Linting et al., 2011*). Users of the package should keep in mind that higher numbers of permutations will increase

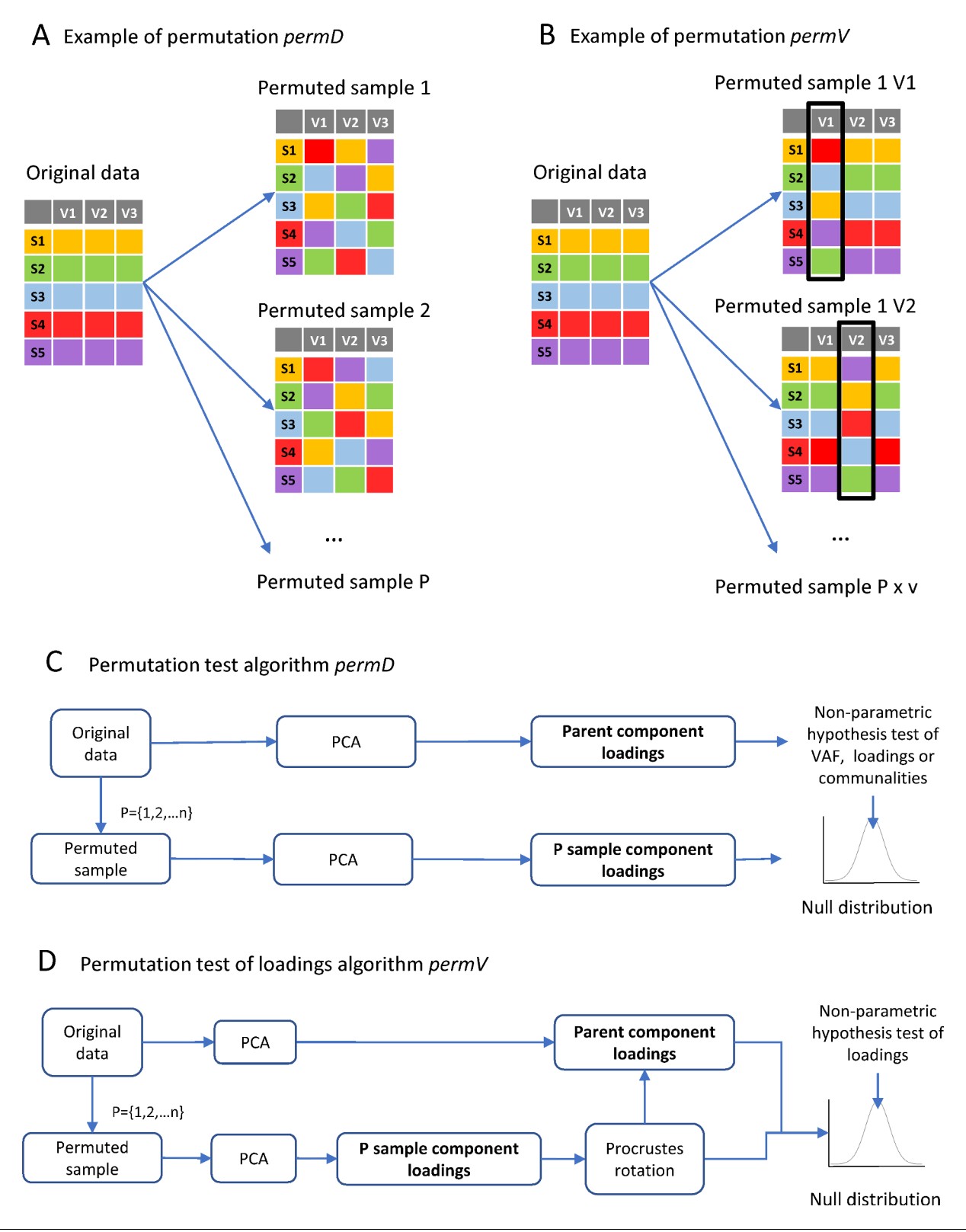

**Figure 2.** Implementation of permutation algorithms. (**A**) Shows a schematic example of the permutation procedure *permD* where all the variables are permuted concomitantly but independently. (**B**) Shows a schematic example of the permutation procedure *permV* where variables are permuted one at the time for each permutation samples (*P*), keeping the other variables as in the original dataset. (**C**) The implemented algorithm for the permutation test algorithm using *permD*: each one to *n* permutation sample (*P*) consist on a random reorganization of observations inside each variable

*Figure 2 continued on next page*

*Figure 2 continued*

independently and concomitantly for each variable. For each *P* sample, a PCA is run and either the loadings, communalities or VAF are calculated. All *P* PCA solutions form the null distribution for non-parametric hypothesis testing of loadings or VAF. (**D**) The permutation test algorithm for loadings under *permV* is performed with and extra step of Procrustes rotation between each of the *P* samples to the parent component loadings. The *P* rotated loadings will then form the null distribution for each variable.

computation time with potentially only a small gain on the approximation. Our simulations indicate that between 500 and 1000 permutations provide a good compromise between computing time and precision in estimating confidence intervals, depending on the data volume (*Figure 4—figure supplement 1*). The package implements a single permutation strategy for VAF, the so-called *permD* (permutation of the entire data set) (*Buja and Eyuboglu, 1992*; *Linting et al., 2011*) where variables are permuted independently and concomitantly (*Figure 2A*) opposed to *permV* (permutation of a single variable) (*Linting et al., 2011*) where variables are permuted one at the time (*Figure 2B*). These methods are further discussed on the component interpretation section.

R Code Box 2.

```
permut_pc_test (pca, pca_data, p=10000, ndim = 5, statistic = 'VAF', perm.method
= 'permD').
```

Case study: After performing a PCA, we first determined the number of components that can be regarded as informative. Several criteria can be used as mentioned earlier. Here, we opted for the permutation test of VAF, computed using the *permut_pc_test()* function (R Code Box 2). We have applied this test to the data using 10,000 permutations. The results show that the three first PCs (PC1, PC2, and PC3) are significantly different from random at an alpha of 0.05 adjusting the p value (*Figure 3*), and therefore we will keep these three PCs for subsequent analysis. PC1 accounts for 32.9% of the variance, PC2 18.3% and PC3 9.8%.

## Step 3: Component interpretation: what do these components mean?

A key part of the analytical workflow is the interpretation of the main components, where the most relevant PCs can be used to represent the correlation between the original variables as a proxy for multivariate disease patterns. Each component is composed of a weighted combination of all the variables. Some components might be explained by only a few variables with high importance, whereas others might have several variables with important contributions to them. There are a few metrics that can be used for interpreting the relation between the original variables and the PCs (*Abdi and Williams, 2010*). In the *syndRomics* package, we use the standardized loadings or correlation vector coefficients (*Jackson and Hearne, 1973*), and the communalities, which are the sum of squared loadings for each variable across selected PCs representing how much of the variance of each variable can be explained by the total number of kept components. Loadings can be interpreted as the Pearson's r correlation coefficient between a PC and a variable, and it is used to assess the contribution of individual variables on each PC and the direction on which the variable moves along the PC (i.e. opposite or same direction as in the interpretation of a correlation). Communalities can be interpreted as the global impact of a variable in the chosen PCA solution.

In general, the strategy consists of determining a threshold for the absolute value of loadings or the communalities above which variables are considered to have important contribution in the definition of a component or across the chosen PCs. For example, if a threshold of |loading| > 0.2 is chosen, all variables for a given PC with a loading > 0.2 or a loading < −0.2 will be considered to contribute on the PC (aka salient variable). The matter then turns to determining an appropriate threshold. Some somewhat arbitrary rules of thumb for the loadings have been established. However, those have a strong determination in psychological studies and whether they are appropriate in biomedical research has yet to be verified. An alternative 'quasi-inferential' method is to use permutation test as discussed above for PC VAF but testing for metrics of variable contribution such as

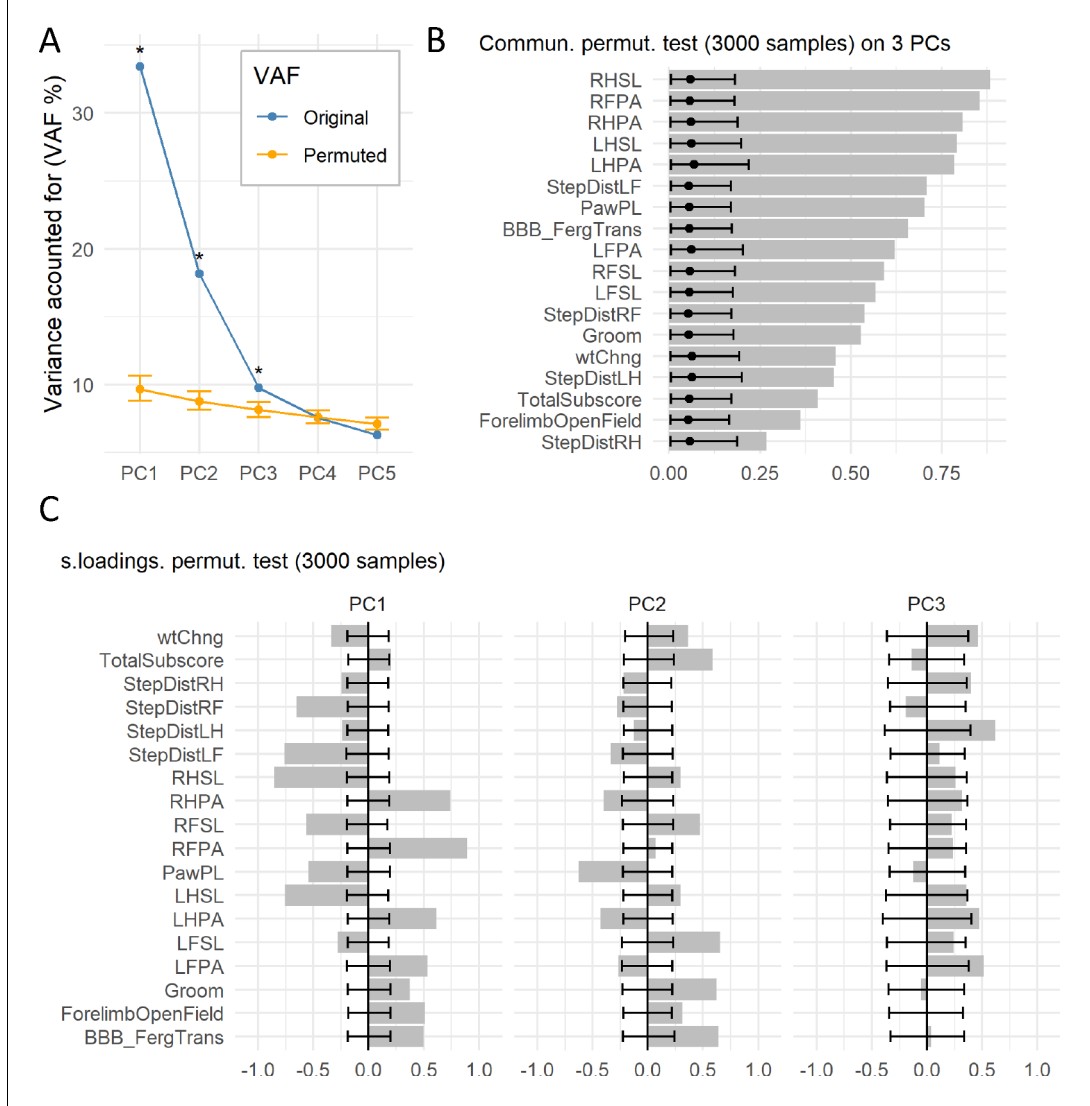

**Figure 3.** Permutation test of case study. (**A**) The graph shows the original VAF for the first five PCs and the average and 95% confidence interval VAF of the permuted PCA distribution (p=10000) using the *permD* method. * Statistical difference for the non-parametric test at alpha = 0.05 and adjusted p value by BH. The three first PCs were selected for the subsequent analysis. (**B**) Barmap of the original communalities (bars) and the permuted distribution (*permV*, p=3000) for each variable calculated over the first three PCs. (**C**) Barmap of the original loadings (bars) and the permuted distribution (*permV*, p=3000) for each variable and each of the first three PCs. Solid dotes represent the mean of the permuted distribution and error bars represent the 95% CI.

The online version of this article includes the following source data and figure supplement(s) for figure 3:

**Source data 1.** csv file containing the source data for panel A in *Figure 3*.

**Source data 2.** csv file containing the source data for panel B in *Figure 3*.

**Source data 3.** csv file containing the source data for panel C in *Figure 3*.

**Figure supplement 1.** Missing data analysis of the first case study.

**Figure supplement 1—source data 1.** csv file for the *Figure 3—figure supplement 1*.

loadings (**Buja and Eyuboglu, 1992**; **Peres-Neto et al., 2003**) or communalities (**Linting et al., 2011**). Using resampling strategies, these permutation methods offer data-driven determination of variable importance and contribution, which might reduce subjective biases. Thus, rejecting the null hypothesis for a given metric, variable and PC, suggest that such variable has a contribution onto the construction of the component that is above what is expected by random noise. As in the case of VAF, this establishes a lower bound for |loadings| or communalities below which they should be

considered noise. In situations with stable solutions and high signal-to-noise ratio, low |loadings| or communalities might still be statistically significant, but the contribution of the variable should be gauged respect to other variables. In the package, we have incorporated permutation test of the loadings as in *Buja and Eyuboglu, 1992*; *Peres-Neto et al., 2003* that can serve to determine the loading threshold, where the variables are permuted independently and concomitantly (*Figure 2A and C*). Linting et al., designed and tested an strategy for the communalities where only one variable is permuted at the time, showing great results in determining the contribution of variables using communalities (*Linting et al., 2011*). This method has resulted in better determination of the significant contribution of variables on the PCA solution with higher statistical power and proper type I error, and therefore has been incorporated in the package as the default method for both the communalities and the loadings (*Figure 2B and D*). Following Linting et al., terminology, users can specify the permutation strategy for the loadings as one variable at the time (*permV*, as in [*Linting et al., 2011*]) or as all the variable together (*permD*, as in [*Buja and Eyuboglu, 1992*; *Linting et al., 2011*; *Peres-Neto et al., 2003*]). See Materials and methods for details on the permutation algorithms. In addition to permutation strategies, the package implements bootstrapping methods for constructing confidence intervals of component loadings and communalities that can also facilitate PCs interpretation (see component stability).

The selection of number of permutations in this case follows similar rationale as described above for the VAF. It is important to note that the minimal number of permutation needed to have enough statistical power and precision will depend on the size of the dataset, both on the number of variables and samples (*Buja and Eyuboglu, 1992*; *Figure 4—figure supplement 1*). There is also the

**Table 2.** List of variables included in the second case study.

| Variable | Description | Values |
|---|---|---|
| CT_Marshall | Marshall CT Score | Range from 1 to 6 |
| CT_Rotterdam | Rotterdam CT Score | Range from 1 to 6 |
| CT_brain_pathology | CT Brain Pathology | 0 = 'No', 1 = 'Yes' |
| CT_skull_FX | CT Skull Fracture | 0 = 'No', 1 = 'Yes' |
| CT_skullbase_FX | CT Skull Base Fracture | 0 = 'No', 1 = 'Yes' |
| CT_facial_FX | CT Facial Fracture | 0 = 'No', 1 = 'Yes' |
| CT_EDH | CT Epidural Hematoma | 0 = 'No', 1 = 'Yes' |
| CT_SDH | CT Subdural Hematoma | 0 = 'No', 1 = 'Yes' |
| CT_SAH | CT Subarachnoid Hemorrhage | 0 = 'No', 1 = 'Yes' |
| CT_contusion | CT Contusion | 0 = 'No', 1 = 'Yes' |
| CT_midlineshift | CT Midline Shift | 0 = 'No', 1 = 'Yes' |
| CT_cisterncomp | CT Cisternal Compression | 0 = 'No', 1 = 'Yes' |
| PTSD_diagnosis_6mo | PTSD DSM-IV Diagnosis (6 months) | 0 = 'No', 1 = 'Yes' |
| GOSE_3mo | GOSE Score (3 months) | Range from 1 to 8 |
| GOSE_6mo | GOSE Score (6 months) | Range from 1 to 8 |
| WAIS_PSI_6mo | WAIS PSI Composite Score (6 months) | Range from 50 to 150 |
| CVLT_short_6mo | CVLT Short Delay Cued Recall Standard Score (6 months) | Range from −4.0–2.5 |
| CVLT_long_6mo | CVLT Long Delay Cued Recall Standard Score (6 months) | Range from −3.5–2.5 |
| SNP_COMT | COMT SNP Genotype | 1 = 'Met/Met', 2 = 'Met/Val', 3 = 'Val/Val' |
| SNP_DRD2 | DRD2 SNP Genotype | 1 = 'C/C', 2 = 'C/T', 3 = 'T/T' |
| SNP_PARP1 | PARP1 SNP Genotype | 1 = 'A/A', 2 = 'A/T', 3 = 'T/T' |
| SNP_ANKK1_Gly318Arg | ANKK1 SNP Gly318Arg | 1 = 'A/A', 2 = 'A/G', 3 = 'G/G' |
| SNP_ANKK1_Gly442Arg | ANKK1 SNP Gly442Arg | 1 = 'C/C', 2 = 'C/G', 3 = 'G/G' |
| SNP_ANKK1_Glu713Lys | ANKK1 SNP Glu713Lys | 1 = 'C/C', 2 = 'C/T', 3 = 'T/T' |

**Table 3.** Communalities of first three PCs on permutation test with 3000 random permutations using *permV* and adjusting p values with BH.

| Variable | Original communalities | Permuted average | Lower 95% CI | Upper 95% CI | p value | Adjusted p value |
|---|---|---|---|---|---|---|
| wtChng | 0.46 | 0.06 | 0.01 | 0.20 | 0.0003 | 0.0004 |
| RFSL | 0.59 | 0.06 | 0.00 | 0.20 | 0.0003 | 0.0004 |
| RFPA | 0.85 | 0.06 | 0.00 | 0.17 | 0.0003 | 0.0004 |
| StepDistRF | 0.54 | 0.05 | 0.00 | 0.17 | 0.0003 | 0.0004 |
| LFSL | 0.57 | 0.06 | 0.00 | 0.18 | 0.0003 | 0.0004 |
| LFPA | 0.61 | 0.06 | 0.00 | 0.20 | 0.0003 | 0.0004 |
| StepDistLF | 0.71 | 0.05 | 0.00 | 0.17 | 0.0003 | 0.0004 |
| RHSL | 0.88 | 0.06 | 0.00 | 0.18 | 0.0003 | 0.0004 |
| RHPA | 0.81 | 0.06 | 0.00 | 0.19 | 0.0003 | 0.0004 |
| StepDistRH | 0.27 | 0.06 | 0.00 | 0.18 | 0.0020 | 0.0020 |
| LHSL | 0.79 | 0.06 | 0.00 | 0.20 | 0.0003 | 0.0004 |
| LHPA | 0.79 | 0.07 | 0.00 | 0.23 | 0.0003 | 0.0004 |
| StepDistLH | 0.46 | 0.06 | 0.00 | 0.20 | 0.0003 | 0.0004 |
| Groom | 0.53 | 0.06 | 0.00 | 0.17 | 0.0003 | 0.0004 |
| PawPL | 0.70 | 0.06 | 0.00 | 0.17 | 0.0003 | 0.0004 |
| BBB_FergTrans | 0.66 | 0.05 | 0.00 | 0.17 | 0.0003 | 0.0004 |
| TotalSubscore | 0.40 | 0.05 | 0.00 | 0.17 | 0.0003 | 0.0004 |
| ForelimbOpenField | 0.37 | 0.05 | 0.00 | 0.18 | 0.0003 | 0.0004 |

understanding that while the *permD* strategy is less robust than *permV* as suggested by Linting et al., the computational time increases considerably since variables are permuted one at the time. Moreover, adjusting p values for multiple testing might be recommended depending on the sample size. Linting et al., suggested controlling for false discovery rate (FDR) using the Benjamini and Hochberg (BH) (*Benjamini and Hochberg, 1995*) method. As a rule of thumb, these researchers advised to only use multiple testing correction (for FDR) when the data contains at least 20 variables and 100 observations or subjects, and to use the uncorrected p-values otherwise (*Linting et al., 2011*). p-Value adjustment has been incorporated in the permutation function on the package, with controlling for FDR by BH as default.

The reader should be cautioned against overinterpreting or misinterpreting the meaning of a PC. The interpretation can be subjective, and unconscious biases can be reflected on the interpretation of PCs. The tools offered by the package help mitigate potential subjective biases, although data biases will affect the results. Another consideration is that it is possible that some of these metrics seem to 'contradict' each other. For example, there is the possibility that a component has an important contribution to the variance of the data (high VAF) and yet all the loadings be small. Contrary, a component with a small set of high loadings could be considered to be insignificant by permuting its VAF (*Buja and Eyuboglu, 1992*). As in any analytical approach, domain knowledge is critical for the interpretation of disease components.

Case study: After deciding to keep three components, we studied the communalities and loadings to determine their identity. Here, we applied the *permut_pc_test()* function (R Code Box 3) setting the argument *statistic* = 'commun' or 's.loadings' and the *perm.method* = 'permV' and using the BH method for controlling for FDR. The results of the permutation test on the communalities can be seen in *Figure 3B* and in *Table 3*. We can appreciate that all variables are significantly represented by the three chosen PCs, although there are five variables with communality less than 0.5, indicating that the retained PCs only explain 50% of the variance on these variables. In PCA, communalities can suggest which variables do or do not contribute to the extracted components altogether. Considering the loadings, the results for PC1, PC2, and PC3 are shown in *Figure 3C* and in *Tables 4*, *5* and *6*, respectively. One can appreciate that the cutoff loading for significance at alpha 0.05 using the adjusted p value is approximately |0.21| for PC1, |0.25| for PC2 and |0.4| for PC3. This

**Table 4.** PC1 loading results of permutation test for the first case study with 3000 random permutations using *permV* and adjusting p values with BH.

| Variable | Original loading | Permuted average | Lower 95% CI | Upper 95% CI | p value | Adjusted p value |
|---|---|---|---|---|---|---|
| wtChng | −0.34 | 0.00 | −0.19 | 0.18 | 0.0003 | 0.0007 |
| TotalSubscore | −0.56 | 0.00 | −0.21 | 0.19 | 0.0003 | 0.0007 |
| StepDistRH | 0.89 | 0.01 | −0.20 | 0.21 | 0.0003 | 0.0007 |
| StepDistRF | −0.65 | 0.00 | −0.19 | 0.17 | 0.0003 | 0.0007 |
| StepDistLH | −0.28 | 0.00 | −0.18 | 0.18 | 0.0043 | 0.0084 |
| StepDistLF | 0.54 | 0.01 | −0.18 | 0.18 | 0.0003 | 0.0007 |
| RHSL | −0.76 | 0.00 | −0.19 | 0.19 | 0.0003 | 0.0007 |
| RHPA | −0.85 | 0.00 | −0.17 | 0.19 | 0.0003 | 0.0007 |
| RFSL | 0.74 | 0.01 | −0.18 | 0.19 | 0.0003 | 0.0007 |
| RFPA | −0.25 | 0.00 | −0.18 | 0.18 | 0.0063 | 0.0114 |
| PawPL | −0.76 | 0.00 | −0.17 | 0.17 | 0.0003 | 0.0007 |
| LHSL | 0.62 | 0.00 | −0.18 | 0.18 | 0.0003 | 0.0007 |
| LHPA | −0.24 | 0.00 | −0.19 | 0.17 | 0.0163 | 0.0259 |
| LFSL | 0.38 | 0.01 | −0.17 | 0.20 | 0.0003 | 0.0007 |
| LFPA | −0.54 | −0.01 | −0.21 | 0.20 | 0.0003 | 0.0007 |
| Groom | 0.49 | 0.00 | −0.20 | 0.19 | 0.0003 | 0.0007 |
| ForelimbOpenField | 0.20 | −0.01 | −0.19 | 0.18 | 0.0323 | 0.0459 |
| BBB_FergTrans | 0.51 | 0.01 | −0.18 | 0.19 | 0.0003 | 0.0007 |

behavior of different thresholds for significance has been previously described (*Buja and Eyuboglu, 1992*) and reflects the fact that PCs accounting for less variance might contain more random noise, thus needing a higher loading for a variable to be considered as an important contributor. Loadings are indicative of both strength of association between a variable and a PC and the direction in which they interact. For reading on the interpretation of the loadings and components in this case study, see *Ferguson et al., 2013*.

R Code Box 3

```
permut_pc_test (pca, pca_data, p=1000, ndim = 3, statistic = 's.loadings', perm.
method = 'permV').
permut_pc_test (pca, pca_data, p=1000, ndim = 3, statistic = 'communa', perm.
method = 'permV').
```

## Step 4: Component stability: how robust are the components?

The presence of a syndrome or disease pattern, represented by a component, should hold true regardless of variations in experiments or metrics meant to measure that same pattern. For example, two experiments with different subjects but the same collected variables should result in inferentially equivalent components if they are true features of the disease and not experimental artifacts. The sensitivity of PCs to experimental, metric, or other forms of variation is termed 'component stability'. Components from different PCAs (from different experiments as an example) that are extremely similar are considered to be a stable, and characterizing component stability is important to determine the robustness of the initial PCA (*Guadagnoli and Velicer, 1988*; *Linting, 2007*). A robust PC would be largely unaffected by data variations (i.e. low sensitivity). The goal of the stability analysis is to determine such sensitivity.

Given that performing multiple replication experiments in biomedical research is not always possible, component stability can be approximated by resampling techniques such as bootstrapping (*Babamoradi et al., 2013*; *Linting et al., 2007a*; *Timmerman et al., 2007*; *Zientek and Thompson, 2007*). Bootstrap methods for component stability have been extensively studied, but users should

**Table 5.** PC2 loading results of permutation test for the first case study with 3000 random permutations using *permV* and adjusting p values with BH.

| Variable | Original loading | Permuted average | Lower 95% CI | Upper 95% CI | p value | Adjusted p value |
|---|---|---|---|---|---|---|
| wtChng | −0.37 | 0.00 | −0.23 | 0.22 | 0.0023 | 0.0047 |
| TotalSubscore | −0.48 | −0.01 | −0.23 | 0.21 | 0.0003 | 0.0007 |
| StepDistRH | −0.07 | −0.01 | −0.23 | 0.23 | 0.5122 | 0.5644 |
| StepDistRF | 0.28 | 0.00 | −0.23 | 0.21 | 0.0143 | 0.0234 |
| StepDistLH | −0.66 | 0.01 | −0.23 | 0.24 | 0.0003 | 0.0007 |
| StepDistLF | 0.27 | 0.00 | −0.22 | 0.23 | 0.0203 | 0.0305 |
| RHSL | 0.34 | 0.00 | −0.24 | 0.23 | 0.0003 | 0.0007 |
| RHPA | −0.30 | 0.00 | −0.21 | 0.21 | 0.0083 | 0.0145 |
| RFSL | 0.40 | 0.00 | −0.22 | 0.25 | 0.0003 | 0.0007 |
| RFPA | 0.21 | 0.00 | −0.22 | 0.22 | 0.0643 | 0.0868 |
| PawPL | −0.30 | 0.00 | −0.19 | 0.22 | 0.0023 | 0.0047 |
| LHSL | 0.42 | 0.01 | −0.21 | 0.23 | 0.0003 | 0.0007 |
| LHPA | 0.12 | 0.00 | −0.24 | 0.22 | 0.3182 | 0.3656 |
| LFSL | −0.62 | 0.00 | −0.25 | 0.25 | 0.0003 | 0.0007 |
| LFPA | 0.63 | 0.00 | −0.24 | 0.26 | 0.0003 | 0.0007 |
| Groom | −0.65 | 0.00 | −0.22 | 0.23 | 0.0003 | 0.0007 |
| ForelimbOpenField | −0.59 | −0.01 | −0.25 | 0.23 | 0.0003 | 0.0007 |
| BBB_FergTrans | −0.32 | −0.01 | −0.22 | 0.22 | 0.0023 | 0.0047 |

be aware of the limitations and advantages of these methods and their performance for component stability depending on the use case (*Babamoradi et al., 2013*; *Guadagnoli and Velicer, 1988*; *Linting et al., 2007b*; *Timmerman et al., 2007*; *Zientek and Thompson, 2007*).

The package implements functionalities to help study the component stability affected by data selection variability by implementing bootstrapping methods (*Babamoradi et al., 2013*; *Linting et al., 2007b*; *Timmerman et al., 2007*; *Zientek and Thompson, 2007*) and stability metrics. The default method used in the package is the simple or ordinary bootstrap consisting of generating a new sample that has the same size (i.e. same number of subjects or observations) and same variables as the original data, but where the subjects have been randomly selected from the data with replacement (*Figure 4A*). This process is repeated several times (here referred as b times) to generate a sample of bootstrapped data. In the first case example, each of the b bootstrapped samples contain 159 subjects and 18 variables, but one subject might appear more than once and another subject might not show up in a specific sample.

Component stability can be studied at the whole component level or at the level of the individual variables through the loadings and communalities. The package implements component similarity indexes (aka factor matching indexes)(*Cattell and Baggaley, 1960*; *Cattell et al., 1969*; *Guadagnoli and Velicer, 1991*) as metrics to study the stability of PCs. These metrics can be used to determine the similarity between the different bootstrapped samples, to test the validity of the extracted component under two or more experimental conditions, to assess the multidimensional equivalence of two or more replication experiments, or to determine the impact of imputing missing values.

Case study: To understand the sensitivity of variables and components to experimental variations, we used the *pc_stability()* function with b = 1000 bootstrapped samples (R Code Box 4), setting the *sim* argument to 'balanced' to perform balanced bootstrapping. The function will return the average of the loadings and the specified similarity metric across all the b samples as well as the specified confidence interval. For this example, the 95% CI (accelerated and bias-corrected, see Materials and methods) and the bootstrapped average can be seen in *Figure 5*. In general, the original loadings are close to the bootstrapped average which indicates that the results are unbiased. Moreover, the confidence regions for most higher value loadings are reasonably small, suggesting

**Table 6.** PC3 loading results of permutation test for the first case study with 3000 random permutations using *permV* and adjusting p values with BH.

| Variable | Original loading | Permuted average | Lower 95% CI | Upper 95% CI | p value | Adjusted p value |
|---|---|---|---|---|---|---|
| wtChng | 0.46 | 0.00 | −0.43 | 0.41 | 0.0183 | 0.0283 |
| TotalSubscore | 0.22 | 0.00 | −0.32 | 0.34 | 0.2463 | 0.2955 |
| StepDistRH | 0.23 | 0.01 | −0.32 | 0.35 | 0.2303 | 0.2826 |
| StepDistRF | −0.19 | 0.01 | −0.32 | 0.34 | 0.3102 | 0.3642 |
| StepDistLH | 0.23 | 0.00 | −0.34 | 0.36 | 0.2083 | 0.2615 |
| StepDistLF | 0.50 | −0.01 | −0.36 | 0.40 | 0.0063 | 0.0114 |
| RHSL | 0.12 | 0.00 | −0.35 | 0.33 | 0.5382 | 0.5698 |
| RHPA | 0.26 | 0.00 | −0.35 | 0.35 | 0.1903 | 0.2446 |
| RFSL | 0.32 | 0.00 | −0.36 | 0.41 | 0.1043 | 0.1374 |
| RFPA | 0.41 | 0.00 | −0.34 | 0.35 | 0.0223 | 0.0326 |
| PawPL | 0.35 | −0.01 | −0.37 | 0.35 | 0.0583 | 0.0807 |
| LHSL | 0.48 | 0.01 | −0.39 | 0.44 | 0.0123 | 0.0208 |
| LHPA | 0.62 | 0.00 | −0.38 | 0.35 | 0.0003 | 0.0007 |
| LFSL | −0.05 | 0.00 | −0.35 | 0.33 | 0.8001 | 0.8308 |
| LFPA | −0.12 | −0.01 | −0.32 | 0.34 | 0.5302 | 0.5698 |
| Groom | 0.03 | 0.01 | −0.34 | 0.34 | 0.8680 | 0.8844 |
| ForelimbOpenField | −0.14 | 0.01 | −0.32 | 0.34 | 0.4802 | 0.5402 |
| BBB_FergTrans | 0.00 | 0.02 | −0.33 | 0.33 | 0.9900 | 0.9900 |

that these loadings are stable to experimental variation. In addition, the similarity metrics for the three PCs suggest component stability, meaning that the composition of the components is also stable. The accepted values for these metrics indicating stability might vary by field and the metric of interest. Some indicative values are mentioned in the respective method section, but the user should be aware of subjective biases when determining a threshold for considering stability (*Lorenzo-Seva and ten Berge, 2006*). Finally, the function will return the average and percentile CI of the communalities, which can be used to assess which variables are more stable in the selected PCA solution.

R Code Box 4.

```
pc_stability (pca, pca_data, B = 1000, ndim = 3, s_cut_off = 0.1, test_similarity =
T, similarity_metric = 'all', sim = 'balanced', barmap_plot = T).
```

Assessing the impact of imputation methods on introducing noise when dealing with missing data should be considered. As described earlier, we used multiple imputation to generate the dataset for analysis. Multiple imputation generates *m* complete datasets where the imputed values might vary, but the observed values are the same (in the case study *m* = 50). We used the stability analysis described above to determine the sensitivity of the PCA solution to variations introduced by imputing missing values. We calculated the similarity metrics between all the 50 imputed datasets for the first three PCs, as well as the loadings. We observed high similarities between the PCs obtained from the imputed datasets (*Table 7*) and the loadings showed narrow CIs (*Figure 3—figure supplement 1*). We concluded that multiple imputation has produced stable solutions with acceptable impact on both the component and variables. A future version of the package might include more robust methods for pooling and testing multiple imputation in PCA context (*van Ginkel and*

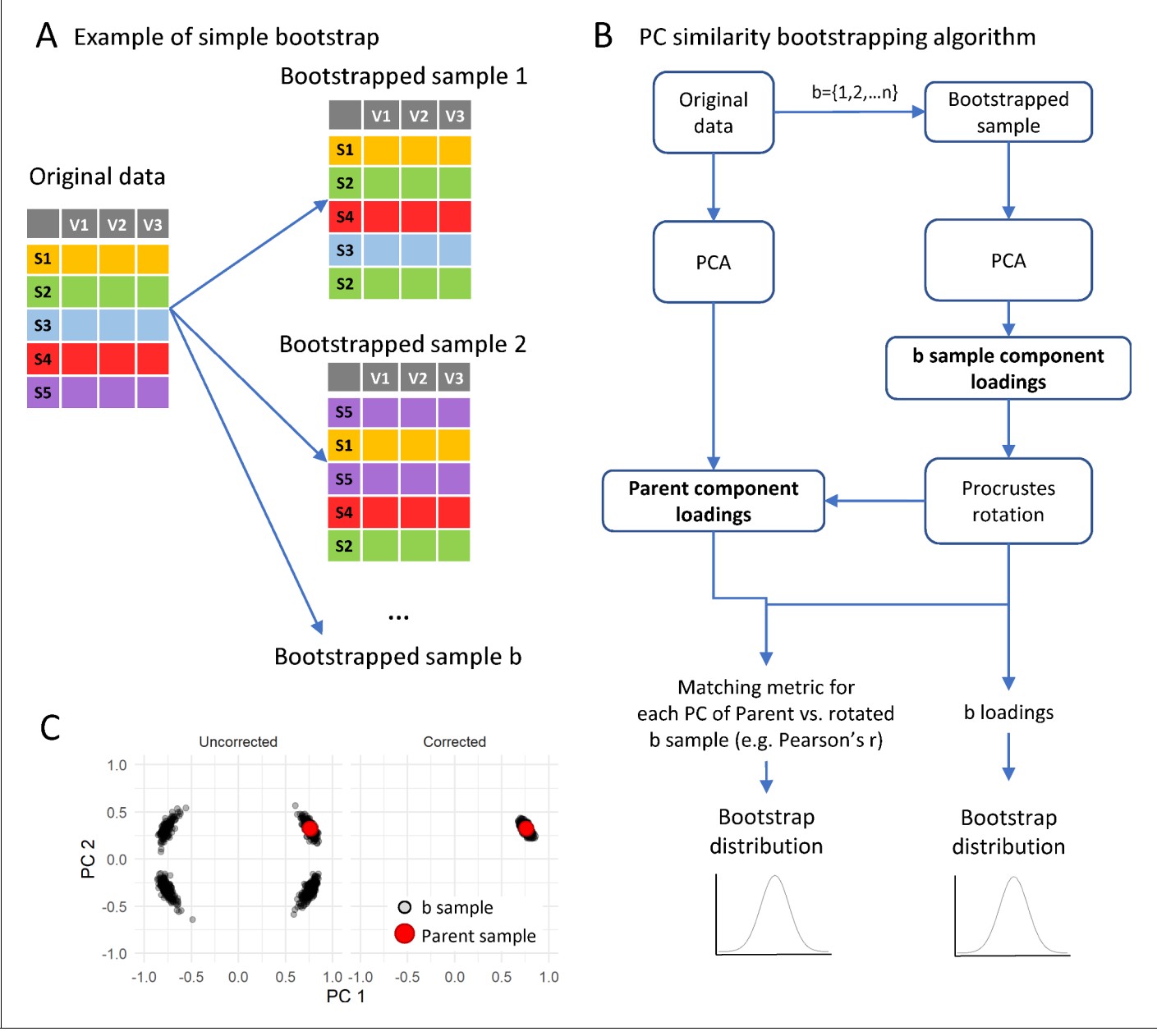

**Figure 4.** Implementation of bootstrapping algorithm. (**A**) shows a schematic of the bootstrapping procedure where a bootstrap sample is generated by resampling the original samples as many times as there are samples in the original dataset but allowing for replacement. The bootstrapping algorithm for loadings is (**B**): for each of 1 to n bootstrap sample (b), run a PCA with the same specifications than the parent PCA on the original sample. The bootstrapping method (e.g. balanced bootstrap) can be specified with the *sim* argument passed to the *boot()* function of the boot R package. Then, the sample component loading is obtained from the PCA of the bootstrapped sample and a Procrustes rotation of the loading matrix is applied over the parent loading matrix to correct for PCA indeterminacies (**C**; see text). All *b* rotated loadings form the bootstrapped distribution of loadings. The component similarity of each *b* loading with the parent loading solution can be calculated to generate the bootstrapped distribution of component similarity. From these distributions, the average and confidence interval are estimated.

The online version of this article includes the following figure supplement(s) for figure 4:

**Figure supplement 1.** Computation time for resampling.

*Kroonenberg, 2014*). Altogether, the results suggest reliable and robust PCs extracted from the original data.

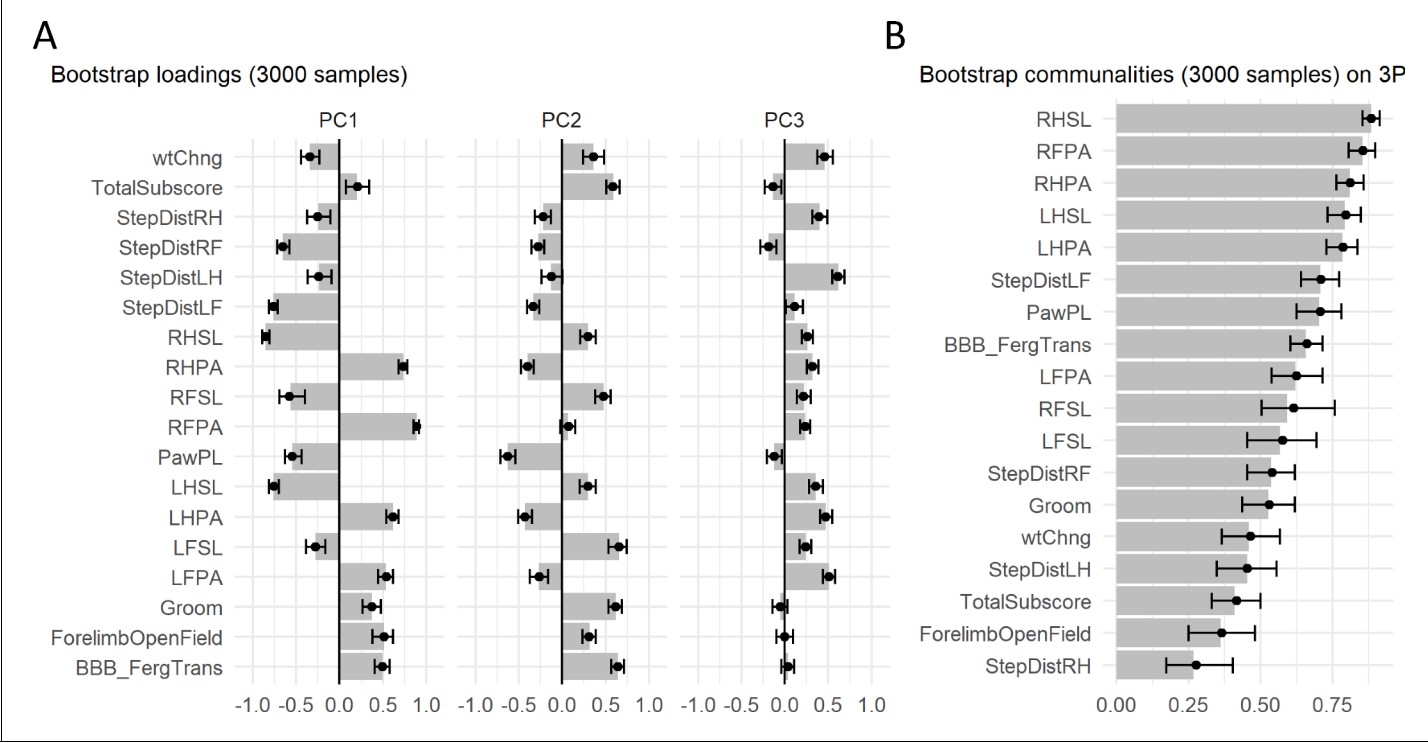

**Figure 5.** Principal component (PC) stability results of case study. Barmap plot of the bootstrap distribution of loadings (**A**) and communalities (**B**) representing the average and the 95% confidence interval of 3000 bootstrapped samples for the first three PCs. Assessing the confidence region offers an indicator of the uncertainty of the estimated loadings for each variable on each PC. Solid dots represent the mean of the bootstrap distribution and error bars represent the 95% CI.

## Step 5: Component visualization

Communicating the analysis is a necessary part of the workflow. Although we have included this at the end of the use case, visualization can be also used for aiding in component selection, component interpretation and component stability analysis. There are several ways a PCA solution can be visualized. Here, we describe the plots implemented in the *syndRomics* package.

We have coded three types of plots (syndromics plot, heatmap, and barmap) using the grammar of the graphics framework (*Wilkinson, 2005*) implemented in R by the *ggplot2* package. This allow users to customize the plots using the rich landscape of the *ggplot2* universe. The syndromic plot was first published by *Ferguson et al., 2013* and represents PCs as the center of a Venn diagram (*Figure 1A*), consisting of (1) a middle convex triangle displaying the 'variance accounted for' (VAF) for a given PC and (2) radial arrows pointing to the center of the triangle for each variable with a standardized loading above a certain threshold (*Figure 6A–C*). The width of each arrow and the color saturation are proportional to the magnitude of the standardized loading they represent. The color of each arrow additionally differentiates between positive or negative loadings (e.g. blue represents a loading of +1, red represents a loading of −1, and white represents a loading of 0).

**Table 7.** Similarity metrics of the first three PCs between 50 multiple imputed datasets for the first case study. Silent cutoff for S index was set at |0.2|.

|     | CC index |        | r index |        | RMSE  |       | S index |       |
| --- | -------- | ------ | ------- | ------ | ----- | ----- | ------- | ----- |
| PC  | Mean     | SD     | Mean    | SD     | Mean  | SD    | Mean    | SD    |
| PC1 | 0.999    | 0.0003 | 0.999   | 0.0003 | 0.021 | 0.004 | 0.991   | 0.015 |
| PC2 | 0.998    | 0.0005 | 0.998   | 0.0005 | 0.021 | 0.004 | 0.93    | 0.042 |
| PC3 | 0.997    | 0.001  | 0.996   | 0.002  | 0.022 | 0.005 | 0.965   | 0.03  |

Syndromic plots are especially useful for conveying PC identity in an easy to understand, concise way for publication. Heatmap and barmap plots are alternative visualizations of the loadings beyond the syndromic plot. The major difference between these two plots and the syndromic plot is that both the barmap (*Figure 5A*) and heatmap (*Figure 6D*) plots display all variables (or a manually selected subset) instead of only the ones with loadings above a given threshold. The absolute loadings that exceed a cutoff threshold can be noted (e.g. by a star *). Moreover, in the case of barmap plots, the cutoff is represented in the graph by vertical lines. This is particularly useful when there are too many above-threshold variables, which would crowd the syndromic plot visualization, or when comparing loadings between PCs more easily. In addition, barmaps are useful for documenting the results of the resampling procedures since error bars can be used to represent the variation of the metrics over the resampling. The *permut_pc_test*() and *pc_stability*() functions return such plots.

Case study (R Code Box 5): We can visualize the three selected PCs using the plotting functions in *syndRomics* (*Figure 6*). In this case, we chose to represent PC1, PC2 and PC3 using the syndromics plots (*Figure 6A, B and C*, respectably) using a cutoff threshold of |0.45|. Notice that this is higher than the threshold for significance found by the permutation analysis given the high number of variables. The full loading pattern of the three first PCs can be visualized by a heatmap (*Figure 6D*), where we have chosen a different cutoff for each PC (0.21, 0.25, and 0.4 for PC1, PC2, and PC3 respectively), or a Barmap (*Figures 3B* and *5A*). Barmaps can be obtained for the loadings (*barmap_loadings()*) or for the communalities (*barmap_commun()*).

```
R Code Box 5
syndromic_plot (pca, pca_data, cutoff = 0.45).
heatmap_loading (pca, pca_data, ndim = 3, cutoff = c(0.21,0.25,0.4), star_values
= T, text_values = F).
```

## Case study 2

In the second case study, we used selected variables from the Transforming Research And Clinical Knowledge in Traumatic Brain Injury (TRACK-TBI) pilot study (*Yue et al., 2013*) that were analyzed previously and made publicly available (*Nielson et al., 2017*). The released dataset version contains 586 de-identified human subjects who were enrolled in the TRACK-TBI pilot study and the 26 selected variables previously analyzed (*Nielson et al., 2017*). These variables are a subset of brain imaging results, outcome metrics and genetic polymorphism (*Table 2*). The goal is to describe patterns of association between these three categories of variables. A noticeable difference between this dataset and the one used in the first case study is that here we are dealing with a mixed type dataset, where some variables are continuous, some nominal and some ordinal. Therefore, we performed a version of nonlinear PCA that allows for the extraction of patterns in these kinds of data. The syndRomics package has been programmed to work with the results of the *princals()* function from the *Gifi* R package. The code for this analysis is found in the supplementary material.

Missing data analysis showed an overall 21.2% missingness distributed between the outcomes and genetic polymorphism variables (*Figure 7—figure supplement 1*). With the exception of 'MRI results' that has high missingness (61.7% of the observations), all imaging variables are complete. 'MRI results' variable was excluded from the analysis. The subsequent test for MCAR suggest that there are 17 different patterns of missingness and that the hypothesis of MCAR can be rejected overall (p-value<0.001). Thus, excluding subjects from the analysis is not justified (*Schafer and Graham, 2002*; *Buuren, 2018*). We instead performed 50 multiple imputations using the *mice* R package as in the first case study. The 50 imputed datasets where then aggregated to perform nonlinear PCA using *princals()* (see Materials and methods for details).

Permutation test of PC VAF suggests that the first 6 PCs contain information that can be regarded as significant above random chance. Although a deep analysis of these six PCs might be of interest, the first three PCs explain the major variance (25.6%, 10.6%, and 9.8%, respectively). Therefore, we focused on interpreting these for illustration purposes (*Figure 7* and *Tables 8–10*). The first PC significantly loaded highly on two genetic variants in opposite directions (SNP_DRD2 loading = −0.677, SNP_ANKK1_Gly318AR loading = 0.661) as well as outcomes of neuropsychological function at 6 months after TBI (CVLT_long loading = −0.614, CVLT_short loading = −0.542)

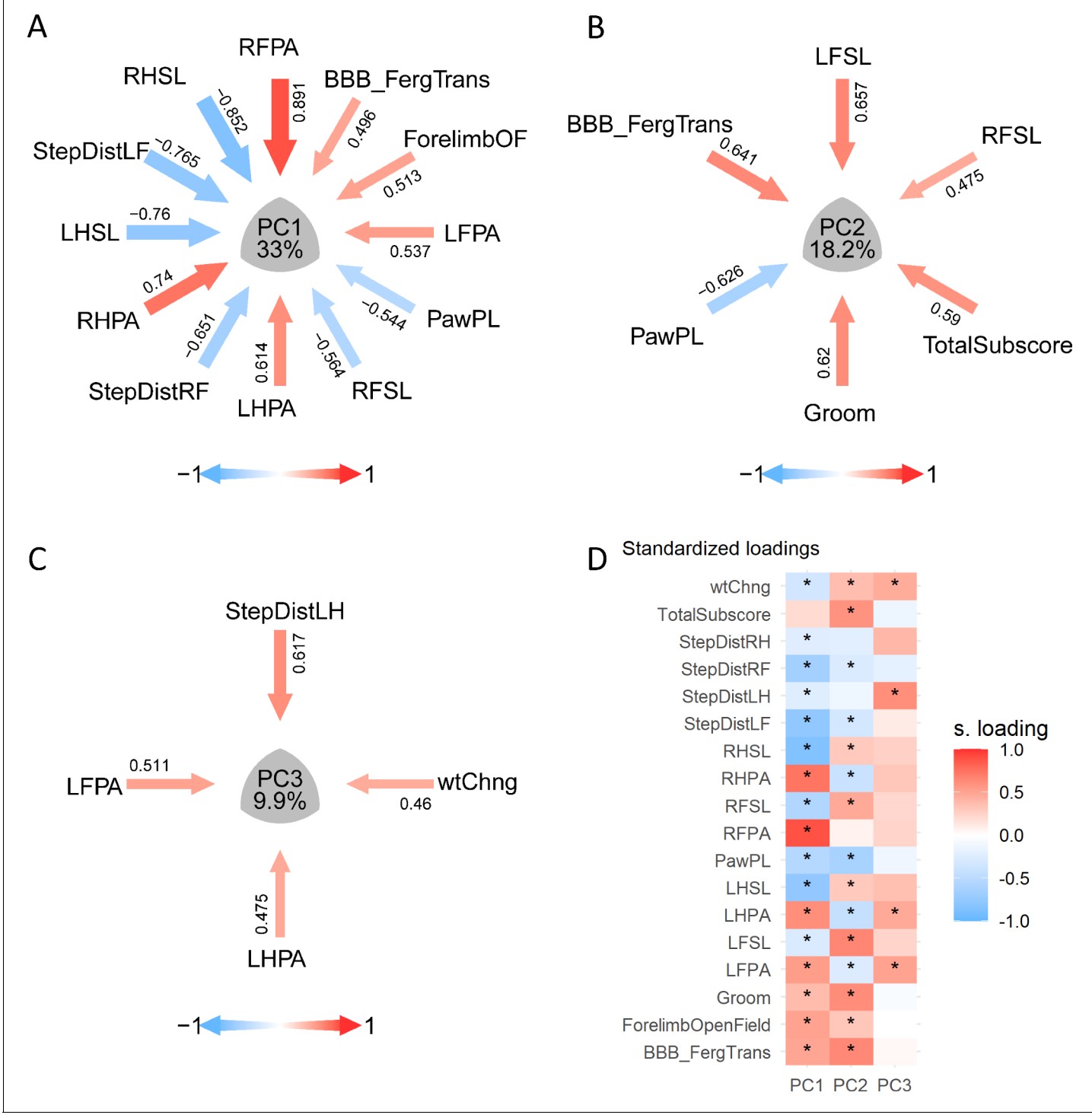

**Figure 6.** Visualization of PCA solutions for syndromic analysis. (A–C) show the layout of the PC1, PC2, and PC3 syndromic plot of variables | loadings| > 0.45, respectively: arrows pointing the center of the plot representing the magnitude (arrow thickness and color saturation) and direction (color) of the loadings of selected variables. (D) illustrate an example of the same loading solution plotted by a heatmap. * Indicates variables with | loadings| > 0.21, 0.25 or 0.4 for PC1, PC2, and PC3, respectively.

(*Figure 7A–C*). All other variables also significantly loaded on to PC1, but with |loadings| ~ 0.3 (*Tables 8–10*) suggesting that their contribution in PC1 identity is less important. Lower values in CVLT (California Verbal Learning Test) suggest learning and memory impairments, which are well

known after TBI. Given the negative loadings for the included CVLT measures (short and long recall), negative values in PC1 might reflect better CVLT outcomes at 6 months after TBI. The stability of the PC1 pattern to multiple imputation is relatively low, with higher loadings showing high variation (*Figure 7—figure supplement 1*, *Table 11*), emphasizing the importance of studying stability of components to multiple imputation. Nonetheless, the bootstrapped loadings were stable (*Figure 7B*), and decay in CVLT performance after TBI has been previously associated to polymorphisms in DRD2 and ANKK1 genes (*Failla et al., 2015*; *McAllister et al., 2008*; *Nielson et al., 2017*; *Yue et al., 2017*), providing literature validation of PC1. The variables with higher positive loadings in PC2 were related to the imaging findings and negative loadings with global function outcomes at 3 and 6 months after TBI (GOSE score)(*Figure 7*. A, D). Lower scores in GOSE are indicative of lower global function and positive values in imaging findings are suggestive of a bigger or more noticeable brain damage. PC2 presented the higher stability to both resampling and to multiple imputation (*Figure 7—figure supplement 1*, *Table 11*). Altogether, PC2 might be interpreted as a surrogate for 'TBI severity', where higher positive values would indicate higher brain damage with less function at 3 and 6 months after injury, a signature described in the previous analysis of this data (*Nielson et al., 2017*). Finally, given the instability of PC3, with most loadings being considered non-significant by the permutation test and the high variance to multiple imputation (*Figure 7—figure supplement 1*, *Table 11*), PC3 can not be interpreted with certainty, and we should not attempt its explanation.

## Discussion

Biomedical research needs more multivariate analytics to help realize the potential of precision medicine. While multiple variables are collected in typical preclinical experiments and clinical trials, univariate statistics continue to be the major analytical and decision-making approaches across the different biomedical fields, narrowing our understanding of the complexity of any disease. With the advent of 'omics', analytical approaches for high-dimensional data have started to become more prevalent for the analysis of biological data. Yet, outside the realm of medical bioinformatics, biomedical research continues to be, for most part univariate. The lack of multivariate approaches in analyzing biomedical data can cause biases and constraints to the interpretation of the results and contribute to the lack of reproducibility and bench-to-bedside translation (*Ferguson et al., 2011*; *Huie et al., 2018*).

The extraction of disease space, through the use of multivariate methods, can increase our understanding of complex relationships commonly present in biomedical data while preventing some of the issues associated with an excessive use of univariate analytics such as multiple comparison testing and associated false discoveries by chance (*Benjamini and Hochberg, 1995*; *Ferguson et al., 2013*; *Krzywinski and Altman, 2014*). For example, it is common in biomedicine to measure several behavioral and histopathological outcomes that are analyzed independently at the univariate level. This approach increases the chance of false-positive results due to the accumulation of type I testing errors (*Benjamini and Hochberg, 1995*; *Dunn, 1961*; *Krzywinski and Altman, 2014*). Although there are methods to correct for errors when running numerous tests such as multiple-testing correction, their use in biomedicine outside of bioinformatic analysis is scarce. Even when correcting for multiple testing, performing several univariate analyses limits our understanding since univariate analysis does not allow us to study and infer the relationship between measures that might capture different aspects of the matter of study. In our first example case study, several functional tests can be used to study the recovery of forelimb motor function after cervical spinal cord injury in animal models. Each test further contains multiple measures about particular aspects of recovery. Knowing the relationship between these measures through multivariate approaches can increase our understanding of the matter of study while reducing the burden of multiple testing (*Ferguson et al., 2013*). Importantly, it is also possible that a single univariate test that does not produce significant results misses true biological effects, while a multivariate analysis including the same variables can find patterns and relationship between variables that are significant. Syndromic analysis is, therefore, a framework that uses multivariate analysis of biomedical data in a holistic way, aiming to reveal interactions within complex (patho)-physiological niches, that would be otherwise challenging to discern. Applying syndromic analysis to biomedical data will help uncover the complex relationships of

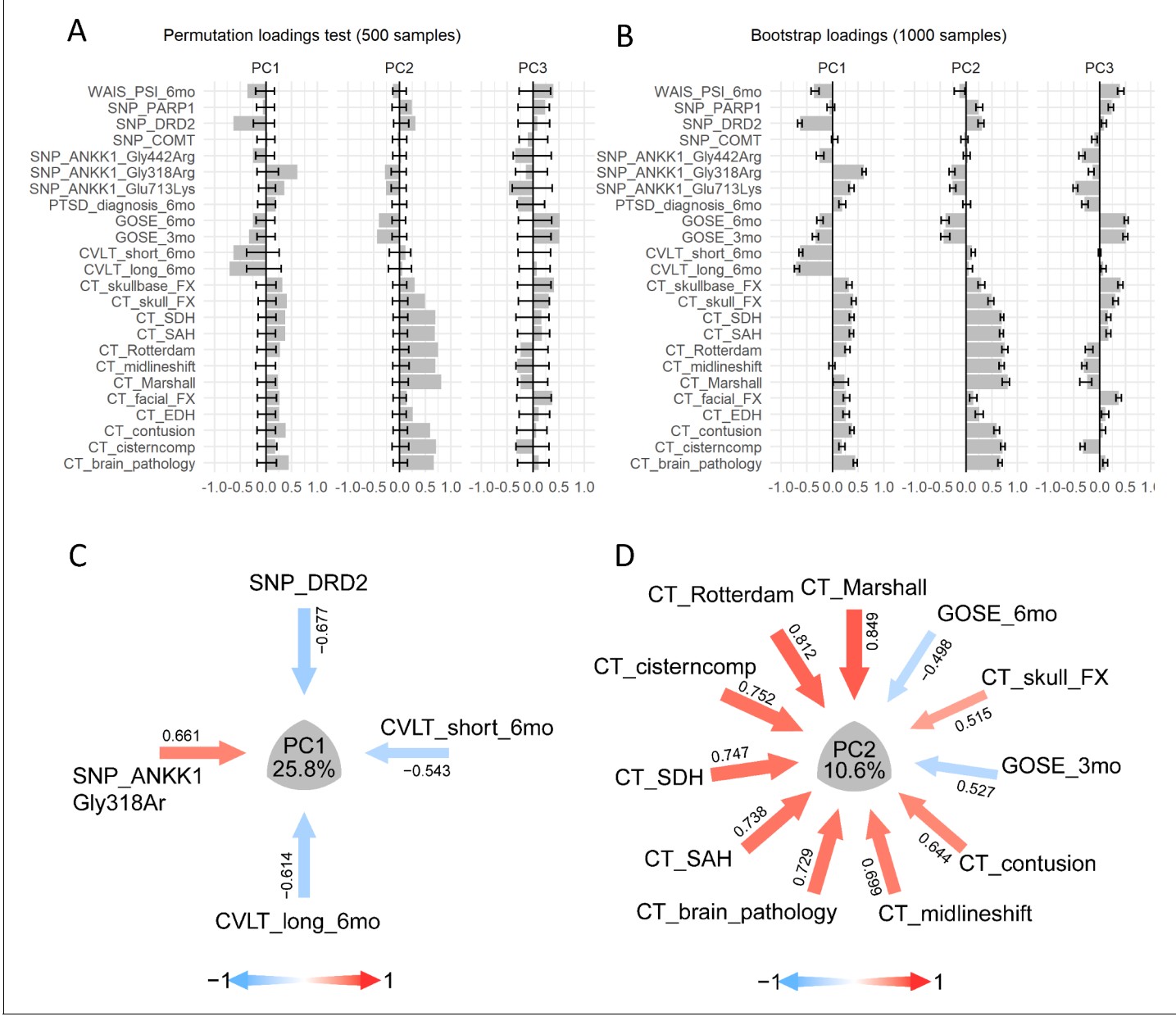

**Figure 7.** Analysis of case study 2 using non-linear PCA and the syndRomics package. (A-B) show thebarmap plots for the loadings for the first three PCs with the 95% CI generated from 500 permutationand 1000 bootstrap resamples. (C-D) show the syndromic plots for the PC1 (VAF=25.8%) and PC2 (VAF=10.6%) for |loading|>0.4. Error bars represent the 95%CI of the resampling method.

The online version of this article includes the following source data and figure supplement(s) for figure 7:

**Source data 1.** csv file containing the source data for *Figure 7*.
**Figure supplement 1.** Missing data analysis of the second case study.
**Figure supplement 1—source data 1.** csv file containing the source data for *Figure 7—figure supplement 1*.

variables and features that constitute different disease and biological states and ultimately accelerate research toward precision medicine.

The *syndRomics* package implements several functionalities for the visualization, the interpretation, and the analysis of the stability of principal components to facilitate the extraction and analysis of disease patterns. We have demonstrated its usage, showing the potential of the package to support PCA-based analysis in understanding disease complexity. Although the core functionalities of the package are included, future versions might also implement outputs from other PCA functions

**Table 8.** PC1 loading results of permutation test for the second case study with 500 random permutations using permV and adjusting p values with BH.

| Variable | Original loading | Permuted average | Lower 95% CI | Upper 95% CI | p value | Adjusted p value |
|---|---|---|---|---|---|---|
| CT_brain_pathology | 0.440589 | 0.010978 | −0.17606 | 0.202095 | 0.001996 | 0.003194 |
| CT_cisterncomp | 0.1844 | 0.003484 | −0.16947 | 0.209801 | 0.053892 | 0.061591 |
| CT_contusion | 0.382612 | 0.008019 | −0.17053 | 0.187363 | 0.001996 | 0.003194 |
| CT_EDH | 0.256368 | 0.015695 | −0.16239 | 0.190167 | 0.001996 | 0.003194 |
| CT_facial_FX | 0.267322 | 0.029918 | −0.16465 | 0.202063 | 0.011976 | 0.015128 |
| CT_Marshall | 0.235242 | 0.003685 | −0.16912 | 0.191229 | 0.00998 | 0.013307 |
| CT_midlineshift | 0.002539 | −0.01082 | −0.2005 | 0.190956 | 0.988024 | 0.988024 |
| CT_Rotterdam | 0.274699 | 0.010723 | −0.17176 | 0.195646 | 0.001996 | 0.003194 |
| CT_SAH | 0.376596 | 0.009848 | −0.16185 | 0.191671 | 0.001996 | 0.003194 |
| CT_SDH | 0.377542 | 0.018196 | −0.15488 | 0.196538 | 0.001996 | 0.003194 |
| CT_skull_FX | 0.404447 | 0.020049 | −0.15398 | 0.198067 | 0.001996 | 0.003194 |
| CT_skullbase_FX | 0.318179 | 0.019653 | −0.18691 | 0.195547 | 0.001996 | 0.003194 |
| CVLT_long_6mo | −0.70622 | −0.05218 | −0.37582 | 0.295769 | 0.001996 | 0.003194 |
| CVLT_short_6mo | −0.63345 | −0.04306 | −0.378 | 0.258602 | 0.001996 | 0.003194 |
| GOSE_3mo | −0.32848 | −0.00894 | −0.17465 | 0.177456 | 0.001996 | 0.003194 |
| GOSE_6mo | −0.25329 | −0.00613 | −0.19298 | 0.176343 | 0.003992 | 0.005988 |
| PTSD_diagnosis_6mo | 0.188743 | 0.013079 | −0.15885 | 0.190364 | 0.041916 | 0.050299 |
| SNP_ANKK1_Glu713Lys | 0.358071 | 0.028077 | −0.14647 | 0.190965 | 0.001996 | 0.003194 |
| SNP_ANKK1_Gly318Arg | 0.613624 | 0.043104 | −0.17747 | 0.244082 | 0.001996 | 0.003194 |
| SNP_ANKK1_Gly442Arg | −0.25194 | −0.02327 | −0.19756 | 0.16577 | 0.005988 | 0.008454 |
| SNP_COMT | 0.036687 | 0.001474 | −0.17511 | 0.172543 | 0.696607 | 0.726894 |
| SNP_DRD2 | −0.62858 | −0.05017 | −0.24105 | 0.168359 | 0.001996 | 0.003194 |
| SNP_PARP1 | −0.05912 | −0.00576 | −0.18101 | 0.166475 | 0.512974 | 0.559608 |
| WAIS_PSI_6mo | −0.36179 | −0.0144 | −0.19646 | 0.168075 | 0.001996 | 0.003194 |

as inputs, such as those from the PCA functions in the *FactoMineR* package (*Lê et al., 2008*) or the *psych* package (*Revelle, 2017*), allowing for better integration to the PCA landscape in R. In addition, other algorithms of bootstrapping and permutation methods for PCA solutions could be incorporated to increase the options and better adapt to the specifics needs of the user (*Hong et al., 2006*; *Linting et al., 2011*; *Vitale et al., 2017*; *Zientek and Thompson, 2007*).

Here, we emphasize guidance and tools for robust determination of PCA-based disease patterns. We have incorporated resampling methods aiming to reduce subjective biases and to study the stability and generality of the analysis. Although we have shown the use of these functions in different contexts along the process, much more work can be done to extend *syndRomics*. For example, we demonstrated the stability of our analysis under multiple imputation, and future research could investigate number of multiple imputations or missing conditions necessary for stable disease pattern detection. In addition, visualization features of syndRomics may be extended to help interpret disease patterns resolved by other multivariate or machine learning tools involving structure coefficients or feature impact scores. The *syndRomics* resampling methods could also be used to estimate the sample size required for stable PCs in the context of syndromic analysis, allowing for sample planning. The implementations in the package are thus positioned to empower both biological and statistical research toward understanding complex biology and diseases.

**Table 9.** PC2 loading results of permutation test for the second case study with 500 random permutations using permV and adjusting p values with BH.

| Variable | Original loading | Permuted average | Lower 95% CI | Upper 95% CI | p value | Adjusted p value |
|---|---|---|---|---|---|---|
| CT_brain_pathology | 0.662378 | 0.021655 | −0.12084 | 0.158839 | 0.001996 | 0.002818 |
| CT_cisterncomp | 0.709989 | 0.024823 | −0.14181 | 0.178126 | 0.001996 | 0.002818 |
| CT_contusion | 0.596311 | 0.01766 | −0.12222 | 0.157785 | 0.001996 | 0.002818 |
| CT_EDH | 0.253079 | 0.002735 | −0.13662 | 0.135348 | 0.001996 | 0.002818 |
| CT_facial_FX | 0.142602 | 0.000971 | −0.12466 | 0.133574 | 0.041916 | 0.055888 |
| CT_Marshall | 0.809847 | 0.026415 | −0.13438 | 0.173771 | 0.001996 | 0.002818 |
| CT_midlineshift | 0.69605 | 0.034813 | −0.12917 | 0.189917 | 0.001996 | 0.002818 |
| CT_Rotterdam | 0.753498 | 0.01539 | −0.13205 | 0.178638 | 0.001996 | 0.002818 |
| CT_SAH | 0.689084 | 0.017617 | −0.12425 | 0.155246 | 0.001996 | 0.002818 |
| CT_SDH | 0.698728 | 0.017798 | −0.12057 | 0.161901 | 0.001996 | 0.002818 |
| CT_skull_FX | 0.493199 | 0.007764 | −0.13921 | 0.161888 | 0.001996 | 0.002818 |
| CT_skullbase_FX | 0.294691 | 0.003769 | −0.12863 | 0.141375 | 0.001996 | 0.002818 |
| CVLT_long_6mo | 0.056095 | 0.019229 | −0.21544 | 0.23499 | 0.674651 | 0.703983 |
| CVLT_short_6mo | 0.115663 | 0.020014 | −0.20152 | 0.215121 | 0.353293 | 0.423952 |
| GOSE_3mo | −0.43692 | −0.00787 | −0.14301 | 0.139508 | 0.001996 | 0.002818 |
| GOSE_6mo | −0.40155 | −0.00849 | −0.14484 | 0.118386 | 0.001996 | 0.002818 |
| PTSD_diagnosis_6mo | 0.004807 | −0.00863 | −0.14863 | 0.140418 | 0.94012 | 0.94012 |
| SNP_ANKK1_Glu713Lys | −0.26204 | −0.01604 | −0.15757 | 0.132164 | 0.001996 | 0.002818 |
| SNP_ANKK1_Gly318Arg | −0.28622 | −0.01954 | −0.15986 | 0.133637 | 0.001996 | 0.002818 |
| SNP_ANKK1_Gly442Arg | 0.033308 | −0.00308 | −0.12662 | 0.129371 | 0.630739 | 0.688078 |
| SNP_COMT | −0.0406 | −0.00094 | −0.14047 | 0.140783 | 0.588822 | 0.67294 |
| SNP_DRD2 | 0.307457 | 0.023806 | −0.11549 | 0.178777 | 0.001996 | 0.002818 |
| SNP_PARP1 | 0.244246 | 0.000876 | −0.14801 | 0.133131 | 0.001996 | 0.002818 |
| WAIS_PSI_6mo | −0.13252 | 0.001279 | −0.13167 | 0.134985 | 0.055888 | 0.070596 |

## Materials and methods

### Availability and requirements

The code to reproduce this analysis can be found in the supplementary material. The data for the first use case comes from the ODC-SCI (Open Data Commons for Spinal Cord Injury, RRID:SCR_016673, http://odc-sci.org), ODC-SCI:26 dataset (https://scicrunch.org/odc-sci/about/odc-sci_26). The data for the second use case comes from TRACK-SCI and can be downloaded from 10.1371/journal.pone.0169490. The package can be installed from GitHub (https://github.com/ucsf-ferguson-lab/syndRomics) where installation instructions, package manual and examples of usage are provided. Descriptions of the arguments and function usage can be found in the internal package documentation once installed or in the package manual. The package has been implemented in R (*R Development Core Team, 2019*) through RStudio (*Team RS, 2018*) using a few other packages beyond the ones bundled in R as dependencies: *dplyr* (*Wickham et al., 2018*), *ggplot2* (*Wickham, 2016*), *stringr* (*Wickham, 2019*), *tidyr* (*Wickham and Henry, 2020*), *ggrepel* (*Slowikowski, 2019*), *ggnewscale* (*Elio Campitelli, 2020*), *pracma* (*Borchers, 2019*), *png* (*Urbanek, 2013*), *boot* (*Canty and Ripley, 2019*; *Davison and Hinkley, 1997*), *rlang* (*Henry and Wickham, 2020*), and *Gifi* (*Mair and Leeuw, 2019*).

**Table 10.** PC3 loading results of permutation test for the second case study with 500 random permutations using permV and adjusting p values with BH.

| Variable | Original loading | Permuted average | Lower 95% CI | Upper 95% CI | p value | Adjusted p value |
|---|---|---|---|---|---|---|
| CT_brain_pathology | 0.110149 | 0.002819 | −0.30242 | 0.309376 | 0.528942 | 0.641916 |
| CT_cisterncomp | −0.32449 | −0.00334 | −0.33379 | 0.305672 | 0.047904 | 0.13839 |
| CT_contusion | 0.060972 | −0.00689 | −0.31444 | 0.270176 | 0.698603 | 0.728977 |
| CT_EDH | 0.104859 | −0.00125 | −0.27352 | 0.321535 | 0.518962 | 0.641916 |
| CT_facial_FX | 0.371649 | 0.062687 | −0.31692 | 0.355691 | 0.01996 | 0.07984 |
| CT_Marshall | −0.24284 | −0.01023 | −0.3038 | 0.288171 | 0.129741 | 0.259481 |
| CT_midlineshift | −0.30562 | −0.01988 | −0.32888 | 0.308785 | 0.063872 | 0.153293 |
| CT_Rotterdam | −0.24002 | −0.01377 | −0.32884 | 0.292959 | 0.161677 | 0.284003 |
| CT_SAH | 0.16969 | −0.0017 | −0.30469 | 0.323896 | 0.347305 | 0.520958 |
| CT_SDH | 0.164146 | 0.011718 | −0.33537 | 0.308091 | 0.339321 | 0.520958 |
| CT_skull_FX | 0.30911 | 0.013422 | −0.28309 | 0.317483 | 0.047904 | 0.13839 |
| CT_skullbase_FX | 0.412507 | 0.027909 | −0.29748 | 0.350075 | 0.005988 | 0.047904 |
| CVLT_long_6mo | 0.071561 | 0.007602 | −0.26795 | 0.330045 | 0.662675 | 0.722918 |
| CVLT_short_6mo | 0.01478 | 0.003403 | −0.29415 | 0.341327 | 0.922156 | 0.922156 |
| GOSE_3mo | 0.512173 | 0.027225 | −0.3035 | 0.347452 | 0.001996 | 0.023952 |
| GOSE_6mo | 0.519654 | 0.030422 | −0.28179 | 0.361045 | 0.001996 | 0.023952 |
| PTSD_diagnosis_6mo | −0.29067 | −0.02728 | −0.31023 | 0.227907 | 0.051896 | 0.13839 |
| SNP_ANKK1_Glu713Lys | −0.47272 | −0.02347 | −0.40038 | 0.368874 | 0.007984 | 0.047904 |
| SNP_ANKK1_Gly318Arg | −0.14092 | −0.02104 | −0.34164 | 0.27799 | 0.379242 | 0.5354 |
| SNP_ANKK1_Gly442Arg | −0.34766 | −0.02656 | −0.38332 | 0.353987 | 0.083832 | 0.182907 |
| SNP_COMT | −0.10171 | 0.001249 | −0.26635 | 0.284647 | 0.53493 | 0.641916 |
| SNP_DRD2 | 0.081296 | 0.013862 | −0.2912 | 0.323114 | 0.61477 | 0.702595 |
| SNP_PARP1 | 0.233007 | 0.010172 | −0.29054 | 0.318385 | 0.165669 | 0.284003 |
| WAIS_PSI_6mo | 0.39706 | 0.017391 | −0.27022 | 0.337713 | 0.011976 | 0.057485 |

## Package implementation

The *syndRomics* package offers two major functionalities for the purpose of aiding in the process of syndromics analysis: (1) visualization functions and (2) functions incorporating resampling methods to determine stability and inference of PCs.

## Visualization functions

The visualization functions are: syndromic_plot(), heatmap_loadings(), barmap_loadings(), barmap_commun() and VAF_plot(). For the visualization functions, the user can pass an R *data.frame* object with the standardized loadings (or other metrics) obtained by running PCA and related multivariate methods in their preferred software. We opted for this approach to avoid requiring specific implementations of PCA. Loadings obtained from any PCA solution can be easily formatted for usage with the *syndRomics* visualization functions. All functions in the package that takes a data frame as argument use the same format (*Table 12*): variables are organized as rows, and the first column is called 'Variables' and contains the names of the respective variables. The other columns contain the PC loadings and are named 'PC1', 'PC2', etc. Alternatively, the visualizations can also be obtained from the output of the *prcomp()* function in the *stats* package in R (linear PCA) or from the output of the *princals()* function in the *Gifi* package in R (non-linear PCA by categorical PCA). Finally, the results from *pc_stability()* and *permut_pc_test()* can be passed to the *plot()* generic function in R as the package incorporate the corresponding S3 method for 'syndromics' class object.

**Table 11.** Similarity metrics of the first 3PCs between 50 multiple imputed datasets for the second case study. Silent cutoff for S index was set at |0.2|.

| | CC index | | r index | | RMSE | | S index | |
|---|---|---|---|---|---|---|---|---|
| PC | Mean | SD | Mean | SD | Mean | SD | Mean | SD |
| PC1 | 0.955 | 0.035 | 0.958 | 0.033 | 0.094 | 0.06 | 0.88 | 0.037 |
| PC2 | 0.992 | 0.004 | 0.991 | 0.0054 | 0.056 | 0.039 | 0.93 | 0.04 |
| PC3 | 0.87 | 0.097 | 0.874 | 0.097 | 0.133 | 0.127 | 0.71 | 0.06 |

### syndromic_plot ()

The list of arguments for the *syndromic_plot()* function are presented in the package manual. The *syndromic_plot()* function will internally call *extract_syndromic_plot()* function (see utility functions) and return a list of *ggplot2* objects containing the syndromic plot for the first *ndim* PCs. For example, if *ndim* = 5, a syndromic plot for PCs 1 to 5 will be generated. Another important argument is the *cut_off*, which determines the threshold of absolute standardized loadings to consider for plotting. This argument is chosen by the user and is required (with no default). Another required argument is *VAF* in case the *syndromic_plot()* function is called using a *data.frame* input. If the output of the *prcomp()* or *princals()* functions is used, the *syndromic_plot()* function extracts *VAF* internally and the user-defined *VAF* will be ignored. When required, *VAF* is a character vector of the form 'XX %","XX%", etc., where XX is the VAF for each PC to plot, starting with the first PC, followed by the second, etc. (e.g. *c('60.1%","25.3%")* for PC1 and PC2, respectively). An issue we found during the implementation is that the arrow visualization does not display correctly in the R graphical device on Windows machines. Rendering the plot into *.pdf format, for instance using the *ggsave()* function from the *ggplot2* package, solves the problem.

### *heatmap_loadings()*, *barmap_loadings()* and *barmap_commun()*

Most of the functionalities described for the *syndromic_plot()* function also apply for the *heatmap_loading()*, the *barmap_loading()*, and the barmap_commun() functions. A noticeable difference in *barmap_loading()* is that the function will plot the PCs specified in *ndim* instead of the first *ndim* components. For example, if *ndim = c(3,4,5)*, components 3, 4, and 5 will be plotted. This allows for more flexibility on which components to plot, such as isolating a single component (e.g. *ndim = 3* will only plot component 3).

### VAF_plot()

This function can be used to plot a VAF plot from a *prcomp()* or *princals()* output. There are two *style* options, 'line' or 'reduced'.

## Resampling functions

There are two major functions using resampling methods, the *permut_pc_test()* function that implements nonparametric permutation test for either PC VAF for aiding in component selection or PC loadings and communalities for aiding in component interpretation, and the *pc_stability()* function that implements bootstrapping of PC loadings for stability analysis. These functions take as input the output of the *prcomp()* or the *princals()* functions in R as well as the original dataset used on these functions as inputs. The specific call of *prcomp()* or the *princals()* used to obtain the original PCA solution is passed down to the resampling functions in the *syndRomics* package, ensuring that the same arguments are used for resampling (with the exception of the *data* argument on the original *prcomp()* or the *princals()* call, that will be internally changed for each resampling iteration).

### permut_pc_test()

In the *syndRomics* package, the null distribution for the permutation test is generated by permuting the values of each variable independently and concomitantly several times (*permD*) or permuting one variable at the time (*permV*) and re-running the PCA on each permuted sample (*Figure 2*; *Buja and Eyuboglu, 1992*; *Glorfeld, 1995*; *Linting et al., 2011*). When *permV* method is

**Table 12.** Template/example of data.frame containing loadings that can be passed to the visualization functions (only the loadings for the first three PCs are shown).

| Variable | PC1 | PC2 | PC3 |
|---|---|---|---|
| wtChng | −0.34 | −0.37 | 0.46 |
| TotalSubscore | −0.56 | −0.48 | 0.22 |
| StepDistRH | 0.89 | −0.07 | 0.23 |
| StepDistRF | −0.65 | 0.28 | −0.19 |
| StepDistLH | −0.28 | −0.66 | 0.23 |
| StepDistLF | 0.54 | 0.27 | 0.50 |
| RHSL | −0.76 | 0.34 | 0.12 |
| RHPA | −0.85 | −0.30 | 0.26 |
| RFSL | 0.74 | 0.40 | 0.32 |
| RFPA | −0.25 | 0.21 | 0.41 |
| PawPL | −0.76 | −0.30 | 0.35 |
| LHSL | 0.62 | 0.42 | 0.48 |
| LHPA | −0.24 | 0.12 | 0.62 |
| LFSL | 0.38 | −0.62 | −0.05 |
| LFPA | −0.54 | 0.63 | −0.12 |
| Groom | 0.49 | −0.65 | 0.03 |
| ForelimbOpenField | 0.20 | −0.59 | −0.14 |
| BBB_FergTrans | 0.51 | −0.32 | −0.03 |

selected to measure the impact of permuting on loadings, a step of Procrustes rotation of each loading matrix toward the original loading matrix is added to resolve sign reflection, rotation indeterminacy and component translocation (*Figure 2D*, see pc_stability for detailed explanation). This step is not performed when the analysis is performed on the communalities since are invariant to such PCA resampling issues (*Linting et al., 2011*). Confidence intervals of the permuted distribution (null distribution) are calculated using the (1-α)x100% (percentile) of the distribution (*Buja and Eyuboglu, 1992*).

The function calls the *permut_pca_D() or permut_pca_V()* utility generic function internally to generate the permuted distribution of the selected metric (either "VAF","s.loadings" or "comuna") using either the *prcomp()* function for linear PCA or the *princals()* function for nonlinear PCA implemented as S3 R method for the class "prcomp" or "princals". If "VAF" is specified the *permD* permutation will be used, ignoring the input of the user on the *perm.method* argument, returning a matrix containing the VAF for the original PCs, as well as the average and the CI of the permuted VAF distribution. In case "s. loadings" or "communa" are specified, the specified permutation method will be considered (i.e. *permD* or *permV*) and the function will return a list of matrices, one for each selected PC, with the original loadings, and the average and CI of the permuted loadings distribution. In both cases, p values are calculated as described in the main text and returned. Adjusted p values using the specified method in the *adjust.method* argument are also returned.

## pc_stability()

Component stability can be studied at the whole component level, known as factor invariance, or at the level of the individual loadings. We have implemented both options in the package. By default, the *pc_stability()* function returns the average and the accelerated and bias-corrected 95% confidence intervals (CI) of the loadings of the bootstrap distribution (*Efron, 1987*). Depending on the sample size and the number of chosen resamples, the bias-corrected CI will fail and the percentile (1-α)x100% CI will be returned (with corresponding notification). In addition, component similarity or factor matching metrics can be computed by setting the *test_similarity*=TRUE, which will call the *component_similarity()* function. For each of the specified similarity metrics, this function returns the

average of the metric and its confidence interval (95% CI by default) by the percentiles of the boot-strap distribution. The confidence level and the CI method for the loadings can be changed by changing the *conf* and *ci_type* arguments. The function uses the *boot()* function for generating the bootstrapped samples and the *boot.ci()* function for extracting the confidence intervals of the load-ings. Both *boot()* and *boot.ci()* are from the *boot* package in R. This allows the use of different boot-strapping strategies such as simple or ordinary bootstrapping (by default) or balanced bootstrapping. The reader is referred to the *boot* package documentation for more details on the different *sim* methods.

A major problem of performing resampling procedures in PCA is what is known as indetermina-cies that can invalidate comparing between bootstrapped samples (*Babamoradi et al., 2013*; *Chan et al., 1999*; *Linting, 2007*; *Timmerman et al., 2007*; *Zabala and Pascual, 2016*). Sign reflec-tion refers to the change of sign on the component loadings in a PC given slight variation of the data. In addition, slight data variation can also cause component/factor translocation, the change in the position of a component in the PCA solution (e.g. PC1 shifts to the position of PC2), especially when two components have similar VAF. Another problem on performing PCAs with variations in the data is the possibility of rotation indeterminacy when the PCA solution of a resampled data presents with a different rotation of the original PCA solution. These issues generate artificially biased bootstrapped distributions, potentially invalidating the procedure (*Timmerman et al., 2007*; *Zientek and Thompson, 2007*). We have implemented a step of procrustes rotation between the original loadings (target) and the bootstrapped sample, as has been previously demonstrated to be a reasonable method to deal with such issues (*Timmerman et al., 2007*; *Zientek and Thompson, 2007*). The Procrustes rotation is obtained by the *procrustes()* function from the *pracma* package. The algorithm for bootstrapping the PCA solutions is represented in *Figure 4* and implemented in the utility function *boot_pca_sample()*. The number of bootstrap samples is set to 1000 by default. The user must be careful on setting the number too low, reducing the performance of the approxi-mation (*Efron, 1987*). However, setting the number of bootstrap samples too high might unneces-sarily increase computing time with little gain (*Figure 4—figure supplement 1*).

## Indexes of component similarity

We have included several component similarity indexes for determining component/factor invariance in *syndRomics*. The function *component_similiarity ()* returns the specified similarity metrics as well as their summary statistics (average and standard deviation, if applicable) from a list of loading matri-ces (*load.list*). The argument *s_cut_off* is used in the calculation of the Cattell's *s* index (see below) and *ndim* is used to limit the number of components from which to compute the indexes from. Each index has been programmed in a separate utility function for convenience. Although they are not meant to be manually called, users can call them to calculate any of these metrics for a given set of two component loadings. The *similarity_metric* argument takes a single character or a vector of char-acters to specify which metrics to compute. These can be: 'cc_index', 'r_correlation', 'rmse' and/or 's_index'. The user can also specify 'all' to get all metrics. Their definitions are documented below.

### Congruence coefficient (CC, 'cc_index')
First suggested by *Burt, 1948*, it was popularized by *Tucker, 1951* and therefore is also known as Tucker's congruence coefficient. It is calculated as (2):

$$\phi_{x,y} = \frac{\sum_{i=1}^{n} x_i y_i}{\sqrt{\sum_{i=1}^{n} x_i^2} \sqrt{\sum_{i=1}^{n} y_i^2}} \tag{2}$$

where $x_i$ and $y_i$ are the loadings of the variable $i$ on the component or factor $x$ and $y$ respectively. $\emptyset(x,y)$ is equivalent to the cosine of the angle between two vectors and is also referred to as the cosine similarity metric. CC is a measure of proportional similarity between two components, and technically the index has a range from -1 (perfect negative congruence) to 1 (perfect positive congru-ence). In practice, because the all the loadings of a PC can be multiplied by -1 without changing the meaning of the PC, the absolute value of CC is considered, which correspondingly ranges from 0 to 1. The closer to 1, the more similar the two components are. Chan et al. discussed the 0.9 rule of

thumb as an indicator of good matching between PCs (*Chan et al., 1999*). The application of CC as a similarity metric for factor invariance has been extensively studied (*Chan et al., 1999*; *Lorenzo-Seva and ten Berge, 2006*).

## Pearson's correlation coefficient (r, 'r_correlation')

The calculation of *r* between two vectors of component loadings has also been used as a pattern matching metric (*Guadagnoli and Velicer, 1991*). It is computed as (3):

$$r_{x,y} = \frac{\sum_{i=1}^{n}(x_i - \bar{x})(y_i - \bar{y})}{\sqrt{\sum_{i=1}^{n}(x_i - \bar{x})^2}\sqrt{\sum_{i=1}^{n}(y_i - \bar{y})^2}} \tag{3}$$

In the *syndRomics* package, the Pearson's correlation coefficient is calculated using the *cor()* function of the *stats* package.

## Root mean square error (RMSE, 'rmse')

RMSE has also been used as a metric for factor matching (*Guadagnoli and Velicer, 1991*). It is calculated as the square root of the average squared difference of the loadings of the variables as (4):

$$RMSE_{x,y} = \sqrt{\frac{\sum_{i=1}^{n}(x_i - y_i)^2}{n}} \tag{4}$$

| | Component 2 | | |
| Component 1 | PS | H | NS |
| --- | --- | --- | --- |
| PS | $f_{11}$ | $f_{12}$ | $f_{13}$ |
| H | $f_{21}$ | $f_{22}$ | $f_{23}$ |
| NS | $f_{31}$ | $f_{32}$ | $f_{33}$ |

where *n* is the number of variables in both components *x* and *y*. A RMSE of 0 determines a perfect matching, and therefore the smaller the RMSE is, the more equivalent the two components *x* and *y* are.

## Cattell's s index ('s_index')

The *s* index was first suggested by *Cattell and Baggaley, 1960*; *Cattell et al., 1969*. It is based on the *factor mandate matrix* (*Cattell and Baggaley, 1960*) where loadings are either one if a component is considered to act on a variable, called a *salient variable*, or 0 if not (forming the *hyperplane* space). Cattell's suggested an arbitrary ± 0.1 cutoff where variables with loadings outside the cutoff range are removed from the *hyperplane* and considered to be *salient variables*. In practice, one might want to alter the threshold depending on the experimental conditions. Any loading inside the cutoff range is then interpreted as having been produced by chance. The s index is calculated from the cross-classification of the common variables of two components/factors:

where PS = positive salient variable; H = hyperplane variable; NS = negative salient variable; $f_{ij}$ is the joint frequency. Positive and negative salient variables are variables outside the cutoff range with positive or negative loadings respectively.

Pattern matching is determined by comparing the cell frequencies in the cross-classification table. Here we implement the simplified form of calculating *s* (5):

$$s = \frac{f_{11} + f_{33} - f_{13} - f_{31}}{f_{11} + f_{33} + f_{13} + f_{31} + \frac{1}{2}(f_{12} + f_{21} + f_{23} + f_{32})} \tag{5}$$

The reader is referred to *Cattell and Baggaley, 1960*; *Cattell et al., 1969*; *Guadagnoli and Velicer, 1991* for details on reasoning and calculations. *s* ranges from 1 (perfect similarity) to −1

(perfect dissimilarity) centered at 0 (pattern due to chance). Similar to CC, the absolute value of $s$ is considered.

## Internal functions

There are internal functions used by the package that the user might never have to call directly, although they are accessible in case the user needs them. Here, we provided a general description of those, leaving the details to the package documentation. All the internal functions to extract similarity metrics are: *extract_cc()*, *extract_s()* and *extract_rmse()*. They all take two numeric vectors and return the corresponding similarity metric between them.

### new_syndromics()

Helper function to construct the 'syndromics' class object that will be use in the S3 generic and method functions. It returns an object of class 'syndromics' of the type list.

### stand_loadings()

This function extracts the standardized loadings from the output of the *prcomp()* or the *princals()* functions. In the case of the *prcomp()* solution, the standardized loadings are calculated as: $s.loadings = eigenvectors \times \sqrt{eigenvalues}$ if the PCA was performed on the standardized (scaled to unit variance) data or $s.loadings = (eigenvector \times \sqrt{eigenvalues})/S$ where $S$ is the vector of the variables standard deviation. In the case of *princals()*, standardized loadings are returned directly in its output and therefore *stand_loadings()* returns those. The function returns a data frame with the standardized loadings in the form of variables as rows and PCs as columns.

### extract_loadings()

This is a wrapper function for *stand_loadings()* with added functionalities such as error breakers that is used by most functions in the package.

### extract_syndromic_plot()

This function is internally called by the *syndromic_plot()* function and returns a *ggplot2* object with the syndromic plot for the specified PC. The only argument that is not present in the *syndromic_plot()* function is the *pc* argument that specifies which PC to plot. Users should always use *syndromic_plot()* function instead of *extract_syndromic_plot()* since *syndromic_plot()* automatically incorporates other functionalities.

### component_similarity()

This function is called by the *pc_stability()* function to calculate the specified similarity metric (see above) between the given list of data frames of loadings. While pc_similarity() uses this function to calculate similarity between the original (parent) loadings and a B sample loadings, the passed list of loadings can be n > 2. Then, the similarity metrics will be calculated between all combinations of n. It returns a list of objects containing a list of the comparisons, a data frame with the averaged metric and the bounds of confidence interval for each specified metric and PC.

### boot_pca_sample()

This generic function is passed to the *statistic* argument of the *boot()* function internally called by the *pc_stability()* function. It implements the bootstrapping algorithm described above (**Figure 2A**). Then the *boot()* function will call *boot_pca_sample()* B times from the specified data and the pca output of the *prcomp()* (through the method *boot_pca_sample.prcomp()*) or *princals()* (through the method *boot_pca_sample.princals()*) function, returning a list of B data frames of loadings. The bootstrapping method can be specified using the *sim* argument.

### permut_pca_D() or permut_pca_V()

This is a generic function internally called by *permut_pca_test()* to produce $P$ permutations of the given output of the *prcomp()* or the *princals()* functions using *permD* or *permV* method. Four S3 R function methods are implemented: *permut_pca_D.prcomp()*, *permut_pca_D.princals()*,

*permut_pca_V.prcomp(), permut_pca_V.princals()*. It returns a list of the results of permuting the data, conducting a PCA and extracting either the VAF or the standardized loadings for each P as in *Figure 2*.

## Plot.syndromics()

This function implement the S3 method for plotting 'syndromics' class objects generated by *pc_stability()* and *permut_pc_test()* functions using the R generic *plot()*. It returns specific plots calling the visualization functions implemented in the package.

## Nonlinear PCA

Nonlinear PCA by optimal scaling and alternating least square was obtained using the *princals()* function from the {Gifi} package in R. We specified to analyze all variables with nominal restriction scaling, allowing for non-monotonic transformations, and set a restriction of 3 degrees in polynomial transformations for nonlinearity. The corresponding instruction was: *princals(nlpca_data, ndim = ncol (nlpca_data), ordinal = FALSE, degrees = 3, knots = knotsGifi(nlpca_data, type='E'))*, where *nlpca_data* is the imputed dataset for case study 2 (see supplementary code for more details).

## Missing data analysis

Details on the code are available as supplementary material. Data wrangling for the two case studies was performed using R packages included in the *Tidyverse* package. Missing pattern visualization were obtained using the *naniar* (*Tierney et al., 2020*) R packages. Test for MCAR was performed using the *TestMCARNormality()* function from the *MissMech* package (*Jamshidian et al., 2014*). Multiple imputation was performed using predicting mean matching method available in the *mice* (*Buuren and Groothuis-Oudshoorn, 2011*) R package, setting the number of imputations to $m = 50$. A list of 50 complete datasets were then obtained and processed by PCA as specified in the main text. For each $m$ dataset, the loadings where extracted and rotated using Procrustes rotation (*pracma* package) toward the average of the imputed datasets. The distributions of loadings and component similarities for the first three PCs where calculated using the *syndRomics* package as described above.

## Acknowledgements

This work is supported by NIH grants NS106899 (ARF), NS088475 (ARF); VA Grants 1I01R $\times$ 002245 (ARF) and I01R $\times$ 002787 (ARF)

## Additional information

### Funding

| Funder | Grant reference number | Author |
|---|---|---|
| National Institutes of Health | NS106899 | Adam R Ferguson |
| National Institutes of Health | NS088475 | Adam R Ferguson |
| Department of Veterans Affairs | I01RX02245 | Adam R Ferguson |
| Department of Veterans Affairs | I01RX002787 | Adam R Ferguson |
| Craig H. Neilsen Foundation | Special Project | Adam R Ferguson |
| Wings for Life | Special Project | Adam R Ferguson |
| Wings for Life | Individual Grant | Abel Torres-Espín |

The funders had no role in study design, data collection and interpretation, or the decision to submit the work for publication.

### Author contributions

Abel Torres-Espín, Conceptualization, Software, Formal analysis, Supervision, Visualization, Methodology, Writing - original draft, Writing - review and editing; Austin Chou, Software, Visualization,

Writing - review and editing; J Russell Huie, Conceptualization, Visualization, Writing - review and editing; Nikos Kyritsis, Software, Validation, Writing - review and editing; Pavan S Upadhyayula, Conceptualization, Software, Writing - review and editing; Adam R Ferguson, Conceptualization, Supervision, Funding acquisition, Visualization, Methodology, Writing - review and editing

### Author ORCIDs
Abel Torres-Espín ⓘ https://orcid.org/0000-0002-9787-8738
Adam R Ferguson ⓘ https://orcid.org/0000-0001-7102-1608

### Decision letter and Author response
Decision letter https://doi.org/10.7554/eLife.61812.sa1
Author response https://doi.org/10.7554/eLife.61812.sa2

## Additional files

### Supplementary files
- Source code 1. The R script reproducing the analysis of this manuscript in a Rmarkdown file.
- Source code 2. A rendered script with code and outputs of running the code in a html file.
- Transparent reporting form

### Data availability
This work used already available data at the Open Data Commons for Spinal Cord Injury (http://odc-sci.org/) and Plos One (https://doi.org/10.1371/journal.pone.0169490).

The following previously published datasets were used:

| Author(s) | Year | Dataset title | Dataset URL | Database and Identifier |
|---|---|---|---|---|
| Ferguson AR, Irvine K-A, Gensel JC, Nielson JL, Lin A, Ly J, Segal MR, Ratan RR, Bresnahan JC, Beattie MS | 2018 | Cervical (C5), unilateral spinal cord injury with diverse 740 injury modalities, multiple behavioral outcomes, and histopathology | https://scicrunch.org/odc-sci/about/odc-sci_26 | Open Data Common for Spinal Cord Injury, (ODC-SCI:26) |
| Nielson JL, Cooper SR, Yue JK, Sorani MD, Inoue T, Yuh EL, Mukherjee P, Petrossian TC, Paquette J, Lum PY, Carlsson GE, Vassar MJ, Lingsma HF, Gordon WA, Valadka AB, Okonkwo DO, Manley GT, Ferguson AR, TRACK-TBI Investigators | 2017 | Uncovering precision phenotype-biomarker associations in traumatic brain injury using topological data analysis | https://journals.plos.org/plosone/article?id=10.1371/journal.pone.0169490 | journal, 10.1371/journal.pone.0169490 |

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
