## [Decision Letter]

**Acceptance summary:**

The paper documents the implementation of syndRomics, an analytical framework for measuring disease states using principal component analysis and multivariate statistics as primary tools for extracting underlying disease patterns in neurological trauma. The method is robust and will serve as an open-source R package for the visualization of disease component more broadly.

**Decision letter after peer review:**

Thank you for submitting your article "Reproducible analysis of disease space via principal components: a brief tutorial and R package (syndRomics)" for consideration by *eLife*. Your article has been reviewed by two peer reviewers, and the evaluation has been overseen by a Reviewing Editor and a Senior Editor. The reviewers have opted to remain anonymous.

The reviewers have discussed the reviews with one another and the Reviewing Editor has drafted this decision to help you prepare a revised submission.

Summary:

The manuscript is a tutorial and an open-source software to analyze disease patterns using principal components analysis. The software – “syndromics” – is available as a part of an R-package. The authors document the implementation of syndromics in the case studies of neurological trauma data and provide a practical guide to the application of PCA to extract disease patterns.

Essential revisions:

This is a well-written manuscript that could be a helpful manual to biomedical researchers in many different fields. The authors present a new software package called "syndRomics", to extract disease features using PCA and describe the reproducible analysis workflow. Whereas there is general enthusiasm, the Editors request that specific issues need to be addressed.

1) If the authors wish to make syndromics as specialized “one-stop” package for neurotrauma data analysis then they should go over and above to show that PCA works on multiple and different data sets than just one that is presented in the manuscript.

---

## [Author Response]

Essential revisions:This is a well-written manuscript that could be a helpful manual to biomedical researchers in many different fields. The authors present a new software package called "syndRomics", to extract disease features using PCA and describe the reproducible analysis workflow. Whereas there is general enthusiasm, the Editors request that specific issues need to be addressed.1) If the authors wish to make syndromics as specialized “one-stop” package for neurotrauma data analysis then they should go over and above to show that PCA works on multiple and different data sets than just one that is presented in the manuscript.

In response to the reviewers, we have incorporated the analysis on another publicly available dataset of clinical neurotrauma as a second case study to illustrate the utility of the proposed analytical workflow and the use of the package. This dataset is a subset of the variables of the Transforming Research and Clinical Knowledge in Traumatic Brain Injury (TRACK-TBI) Pilot Study. The mixed variable type nature of the dataset allows us to illustrate the use of the package in a nonlinear variant of PCA obtained from functions implemented in the Gifi R package. We demonstrate that the use of syndRomics analysis through nonlinear PCA in this dataset resolves in patterns of variable associations previously described in the literature, but in an unsupervised manner, illustrating the value of the analysis for extracting informative disease patterns. Moreover, this second case study serves as an example of the importance of studying the sensitivity of the analysis in different settings such as multiple imputation of missingness. Altogether, we believe the incorporation of this second use case helps to illustrate generalizability of the package and the analytical workflow in different biomedical settings. The analysis has been added at the end of the Results section, including a new figure in the main text (Figure 7), a new supplementary figure (Figure 7—figure supplement 1) and four new tables.